# RANK+TLR2+ myeloid subpopulation converts autoimmune to joint destruction in rheumatoid arthritis

Weixin Zhang[1,2†], Kathleen Noller[3†], Janet Crane[1,4], Mei Wan[1], Xiaojun Wu[5], Patrick Cahan[3], Xu Cao[1,3]*

[1]Department of Orthopaedic Surgery, Johns Hopkins University, Baltimore, United States; [2]Zhejiang Chinese Medical University, Hangzhou, China; [3]Department of Biomedical Engineering, Johns Hopkins University, Baltimore, United States; [4]Division of Pediatric Endocrinology, Johns Hopkins University School of Medicine, Baltimore, United States; [5]Department of Pathology, Johns Hopkins Medical Institutions, Washington, United States

**Abstract** Joint destruction is the major clinic burden in patients with rheumatoid arthritis (RA). It is unclear, though, how this autoimmune disease progresses to the point of deterioration of the joint. Here, we report that in a mouse model of RA the upregulation of TLR2 expression and its α(2,3) sialylation in RANK+ myeloid monocytes mediate the transition from autoimmunity to osteoclast fusion and bone resorption, resulting in joint destruction. The expression of α(2,3) sialyltransferases was significantly increased in RANK+TLR2+ myeloid monocytes, and their inhibition or treatment with a TLR2 inhibitor blocked osteoclast fusion. Notably, analysis of our single-cell RNA-sequencing (scRNA-seq) libraries generated from RA mice revealed a novel RANK+TLR2− a subset that negatively regulated osteoclast fusion. Importantly, the RANK+TLR2+ subset was significantly diminished with the treatments, whereas the RANK+TLR2− subset was expanded. Moreover, the RANK+TLR2− subset could differentiate into a TRAP+ osteoclast lineage, but the resulting cells did not fuse to form osteoclasts. Our scRNA-seq data showed that *Maf* is highly expressed in the RANK+TLR2− subset, and the α(2,3) sialyltransferase inhibitor-induced *Maf* expression in the RANK+TLR2+ subset. The identification of a RANK+TLR2− subset provides a potential explanation for TRAP+ mononuclear cells in bone and their anabolic activity. Further, TLR2 expression and its α(2,3) sialylation in the RANK+ myeloid monocytes could be effective targets to prevent autoimmune-mediated joint destruction.

## Editor's evaluation

New cell populations of RANK+TLR2- cells and RANK+TLR2+ cells have been identified in RA mice by scRNA-Seq method. The RANK+TLR2- cells could differentiate into a TRAP+ osteoclast lineage but did not fuse to osteoclasts, while RANK+ TLR2+ myeloid monocyte population plays an important role in fusion and osteoclast formation. These findings, with other results, provide novel information on progression of RA disease.

## Introduction

RA is one of the most prevalent autoimmune diseases. Knee and finger joint swelling, destruction, and pain are the common clinic symptoms (*Agemura et al., 2022*; *Smolen et al., 2016*). The autoimmune-mediated chronic inflammation during RA progression stimulates excessive osteoclast formation and resorptive activity, leading to the destruction of the endochondral subchondral bone and cartilage of

*For correspondence:
xcao11@jhmi.edu

† Equal contribution authors

the joints (*Hasegawa and Ishii, 2022*). Previous research efforts in RA, which have been extensive, have mostly focused on the molecular and cellular mechanisms involved in this inflammation-induced joint destruction (*Tsukasaki and Takayanagi, 2019*). Even so, the exact mechanisms by which auto-immunity triggers osteoclast formation and thus joint destruction are still not completely understood. In recent years, inhibition of osteoclast formation by biologic disease-modifying antirheumatic drugs (bDMARDs) has shown promise in the management of RA (*Tanaka and Ohira, 2018*). These drugs directly target inflammatory cytokines, such as tumor necrosis factor (TNF)-α and receptor activator of nuclear factor-κ B ligand (RANKL), and clinic trials with these DMARDs, particularly anti-RANKL (Denosumab), have shown some efficacy in the reduction of osteoclast-mediated joint destruction (*Burmester and Pope, 2017*; *Schett and Gravallese, 2012*; *Tanaka and Tanaka, 2021*). But as these cytokines are also important for general immunity, the use of these inhibitors, especially chronically, is not without significant side effects. Therefore, a more detailed understanding of the molecular signaling mechanism that underpins inflammatory-induced osteoclast fusion and resorptive activity could lead to the identification of a more efficacious and safer therapeutic strategy to slow or prevent joint destruction during RA progression.

Osteoclasts are multinucleated giant cells derived from the fusion of monocytes, and they mediate bone resorption. While macrophage colony-stimulating factor (M-CSF) promotes monocyte survival and proliferation, RANKL is responsible for inducing the commitment of tartrate-resistant acid phosphatase-positive (TRAP⁺) mononuclear cells to a preosteoclast fate, while also subsequently promoting their cell–cell fusion to form TRAP⁺ multinuclear osteoclasts (*Baroukh et al., 2000*; *Boyle et al., 2003*). Significant progress has been made in the identification of the mechanism by which autoimmunity-derived inflammation increases the expression of RANKL and the subsequent increase in osteoclast formation during RA progression (*Coury et al., 2019*; *Okamoto et al., 2017*). Specifi-cally, during autoimmunity dendritic cells present autoantigens to T cells, resulting in the generation of different T helper (Th) cells, including Th17 cells. Th17 cells, in turn, stimulate the fibroblast lineage cells to express RANKL, which activates innate immune cells to produce proinflammatory cytokines, such as TNF-α, interleukin (IL)–1, and IL-6, which together promote osteoclast formation (*Tsukasaki and Takayanagi, 2019*). Forkhead box protein 3 (Foxp3) is an indispensable transcription factor for Treg cell development and function (*Komatsu and Takayanagi, 2018*). Loss of Foxp3 expression generates a novel Th17 cell subset, termed exFoxp3Th17 cells (*Komatsu et al., 2014*). These cells also induce osteoclast formation, primarily by up-regulating RANKL expression in mesenchymal stromal/stem cells (MSCs) (*Komatsu and Takayanagi, 2018*).

In adults, bone is under constant remodeling due to microcracks that occur as a result of the stress of gravity that all terrestrial animals experience. As part of the remodeling process osteoclasts resorbs bone, which in turn results in the release from the bone surface of factors, including trans-forming growth factor-β1 (TGF-β1) and insulin-like growth factor I (IGF-I), that recruit MSCs to the local bone environment. The MSCs, in turn, are induced to differentiate into osteoblasts, which are the cells responsible for bone formation. This 'coupling' between osteoclast and osteoblast activities maintains proper bone homeostasis (*Feng and Teitelbaum, 2013*; *Crane et al., 2016*). Aberrant osteoclast formation or uncoupled osteoclast activity leads to many major skeletal disorders (*Schett and Gravallese, 2012*; *Firestein, 2003*; *Rana et al., 2018*), including RA-associated bone destruction.

The downstream signaling molecules involved in osteoclast fusion have been reported to include Dendritic cell-specific transmembrane protein (DC-STAMP), Syncitin-1, OC-stimulatory transmem-brane protein (OC-STAMP), CD9, and CD44/Matrix metallopeptidase 9 (MMP-9) (*Gambari et al., 2020*; *Boyle et al., 2003*). We recently reported that sialylation of TLR2 induces it's binding to Siglec-15 in TRAP⁺ preosteoclasts to initiate cell-cell fusion during osteoclast formation. In particular, we found that RANKL induces the expression of the sialyltransferase ST3GAL1 via c-Fos, leading to the sialylation of TLR2 and thus it's binding to Siglec-15. We further showed that intrafemoral injection of sialidase, to inhibit the sialylation, resulted in reduced osteoclast fusion and bone resorption in mice (*Dou et al., 2022*). Therefore, TRAP⁺ cell recognition mediated by the binding of sialylated TLR2 to Siglec15 initiates cell fusion and mature osteoclast formation.

Sialylation-mediated control of osteoclast fusion provides a unique novel therapeutic target to treat joint destruction in RA. In this study, we sought to investigate whether inhibition of sialylation could reduce osteoclast formation and subsequent joint destruction in a mouse model of RA. Further, analysis of our scRNA-seq libraries revealed that RANK⁺TLR2⁺ and RANK⁺TLR2⁻ myeloid subsets are

associated with the regulation of osteoclast fusion. Along these lines, we found that in RA mice the RANK+TLR2+ subset was significantly expanded relative to control mice, whereas injection of a sialylation inhibitor significantly decreased osteoclast formation and joint destruction and a decrease in the RANK+TLR2+ subset. Thus, targeting this pathway may represent a novel therapeutic avenue in the treatment of RA-associated joint destruction.

## Results

### Expression of RANK and TLR2 peaks in monocytes before they commit to a TRAP+ macrophage lineage

To examine how autoimmunity is transformed into osteoclast joint destruction during RA progression, we prepared scRNA-seq libraries from single-cell suspensions from the bone marrow of CIA mice and vehicle-injected control mice. Quality control was performed to exclude potential doublets and low-quality libraries. We performed clustering (*Traag et al., 2019*) to identify the major transcriptional states in our data (*Figure 1—figure supplement 1A*). Clusters were annotated based on known marker gene expression and differential gene expression, and SingleCellNet (*Tan and Cahan, 2019a*) was used to classify individual cells based on a well-annotated reference dataset (*Tabula Muris Consortium et al., 2018*, *Figure 1—figure supplement 1B*). Once monocytes and macrophages were successfully identified (*Figure 1—figure supplement 1C*), myeloid cells expressing marker genes relevant to our study of osteoclastogenesis—namely RANK (*Tnfrsf11a*), TRAP (*Acp5*), ST3GAL4 (*St3gal4*), and/or TLR2 (*Tlr2*)—were isolated and the data analysis was repeated on those cells (*Figure 1A and B*). Slingshot pseudotime (*Street et al., 2018*) was computed on this population, selecting macrophages as the terminal cluster. Pseudotime values were used to partition the cells into transcriptionally distinct stages, or 'epochs,' and to identify dynamically expressed genes. The expression of *Tlr2* and of two key myeloid/pre-osteoclast marker genes (*Acp5* and *Tnfrsf11a*) were found to significantly change across pseudotime, suggesting that these genes may play a temporal role in the differentiation of monocytes into macrophages in the pre-osteoclast lineage. We visualized the pseudotemporal expression of these genes (*Tnfrsf11a*, *Acp5*, and *Tlr2*) in pre-osteoclasts and found that *Tlr2* expression peaks in the late monocyte phase (*Figure 1B*). This peak in *Tlr2* expression directly precedes a peak in *Tnfrsf11a* expression at the monocyte-to-macrophage transition period (*Figure 1B*), implicating a key role for TLR2 in monocyte differentiation and osteoclastogenesis. Interestingly, a peak in sialyltransferase (*St3gal4*) expression coincides with that of *Tlr2* (*Figure 1B*). We further found that *Acp5* mRNA levels increased following *Tnfrsf11a* expression in the macrophage (*Figure 1B*), validating the known role of RANKL in triggering the differentiation of myeloid cells into TRAP+ mononuclear preosteoclasts (*Boyle et al., 2003*; *Koga et al., 2004*). To validate these characteristics of gene expression, we isolated the primary bone marrow monocytes and cultured them with an osteoclast induction medium (M-CSF 30 ng/ml, RANKL 200 ng/ml). We performed TRAP staining and immunofluorescence staining of RANK and TLR2 at 0, 1, 3, and 5 days of culturing, and we found that the percentage of TRAP+ cells significantly increased following the different time points and come to a peak on day 3 (*Figure 1C–D*). We further found that the number of multinuclear osteoclasts significantly increased from the third day onward, which occurred later than the peak expression of RANK protein and TLR2 protein, which both peaked on day 1 but then subsequently declined (*Figure 1C–G*). This time course of RANK and TLR2 protein expression is consistent with the gene expression profile of the scRNA-seq data. These results show that increased expression of RANK and TRL2 is associated with the commitment of monocytes to a TRAP+ lineage to promote osteoclast formation.

### Expression of osteoclast fusion genes peak during RANK+TLR2+ monocyte commitment to a TRAP+ lineage, but not in RANK+TLR2− monocytes

To characterize the difference between TLR2 positive and negative subpopulations of RANK+ myeloid cells, we further investigated the dynamically expressed genes across pseudotime in these two subpopulations from our scRNA-seq data. The pseudotime was partitioned into two time periods, or 'epochs,' for dynamic gene network reconstruction. The first 'epoch' is comprised solely of monocytes and represents the first stage of the RANK+ myeloid differentiation process, whereas the second 'epoch' contains late monocytes and macrophages. To compare RANK+TLR2+ and RANK+TLR2− myeloid

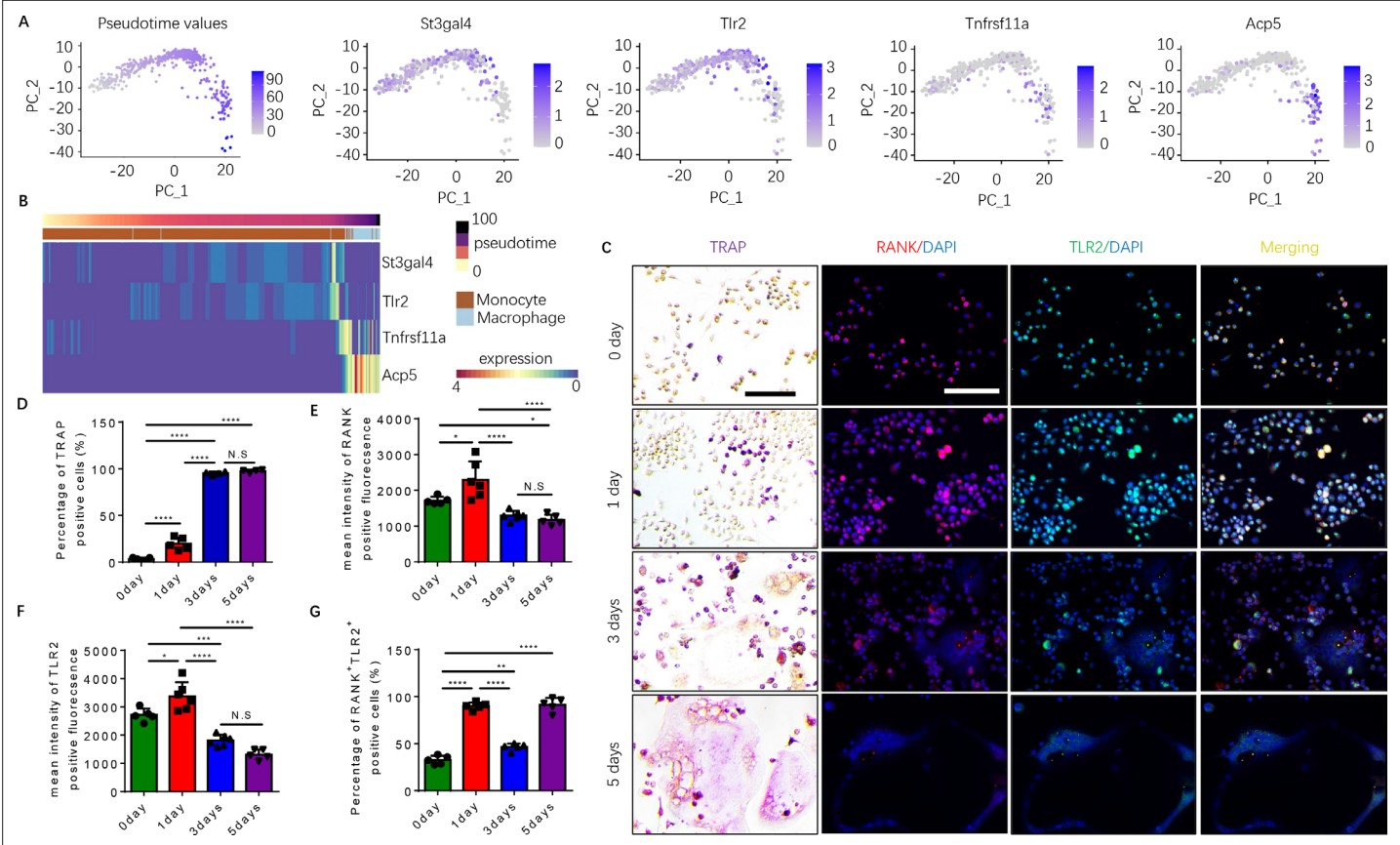

**Figure 1.** Expression of RANK (*Tnfrsf11a*), TLR2 (*Tlr2*), and tartrate-resistant acid phosphatase (TRAP) (*Acp5*) is elevated during the commitment of monocytes to a tartrate-resistant acid phosphatase-positive (TRAP+) lineage. (**A**) Dimensional reduction projection of monocyte and macrophage cells expressing RANK, TRAP, and/or TLR2 from all sample groups onto two dimensions using principal components, depicting values of Slingshot pseudotime followed by gene expression levels of *St3gal4, Tlr2, Tnfrsf11a,* and *Acp5*. (**B**) A heat map of dynamic expression of *St3gal4, Tlr2, Tnfrsf11a,* and *Acp5* in monocyte and macrophage cells expressing RANK, TRAP, and/or TLR2 from all sample groups. Pseudotime values and annotated cell types are depicted in the color-coded bars atop the heat map. (**C**) Representative images of TRAP staining and immunofluorescence co-staining for RANK (red), TLR2 (green), and DAPI (blue) staining for nuclei for bone marrow macrophages cultured with macrophage colony stimulating factor (M-CSF) ( + ng/ml) and RANKL (200 ng/ml) for 0, 1, 3, and 5 days. Scale bar, 0.1 mm. (**D**) Quantitative analysis of the percentage of TRAP+ cells in bone marrow macrophages at different time points (n=4 or 5, one-way ANOVA with Tukey's multiple comparisons test). (**E–G**) Quantitative analysis of the mean intensity of RANK and TLR2 positive fluorescence, and the percentage of RANK+TLR2+ positive cells in bone marrow macrophages at different time points (n=5 or 6, one-way ANOVA with Tukey's multiple comparisons test). All data are means ± SD. N.S=No Significant difference, *p<0.05, **p<0.01, ***p<0.001, ****p<0.0001.

The online version of this article includes the following figure supplement(s) for figure 1:

**Figure supplement 1.** Clustering and annotation of scRNA-seq data.

subpopulations, the dynamic expression of osteoclast fusion-related genes (*Figure 2—figure supplement 1A - B* ) and osteoclast differentiation-related genes (*Figure 2—figure supplement 1C, D*) in each subpopulation were analyzed. In epoch 2, the mean expression of the osteoclast differentiation-related gene *Csf1r* and the osteoclast fusion-related genes *Ccr2* and *Spic* were significantly higher in the RANK+TLR2+ subpopulation as compared to the RANK+TLR2− subpopulation (*Figure 2A*). Furthermore, several osteoclast differentiation marker genes (*Fos* and *Nfkb*) and cell fusion marker genes (*Cd44, Ccr2, Creb3,* and *Clcn3*) were widely expressed in the early stages of RANK+ myeloid cell differentiation, but their expression was markedly reduced at the monocyte-to-macrophage transition of the RANK+TLR2− subpopulation (*Figure 2B*). In contrast, these osteoclast differentiation and cell fusion marker genes were continuously expressed through the differentiation process in the RANK+TLR2+ subpopulation (*Figure 2B*). These results further suggest that TLR2 regulates osteoclast fusion differentiation and that RANK+TLR2− myeloid cells do not undergo a fusion into multinuclear osteoclasts. It should be noted that a subpopulation of periosteal TRAP+ mononuclear cells exists that

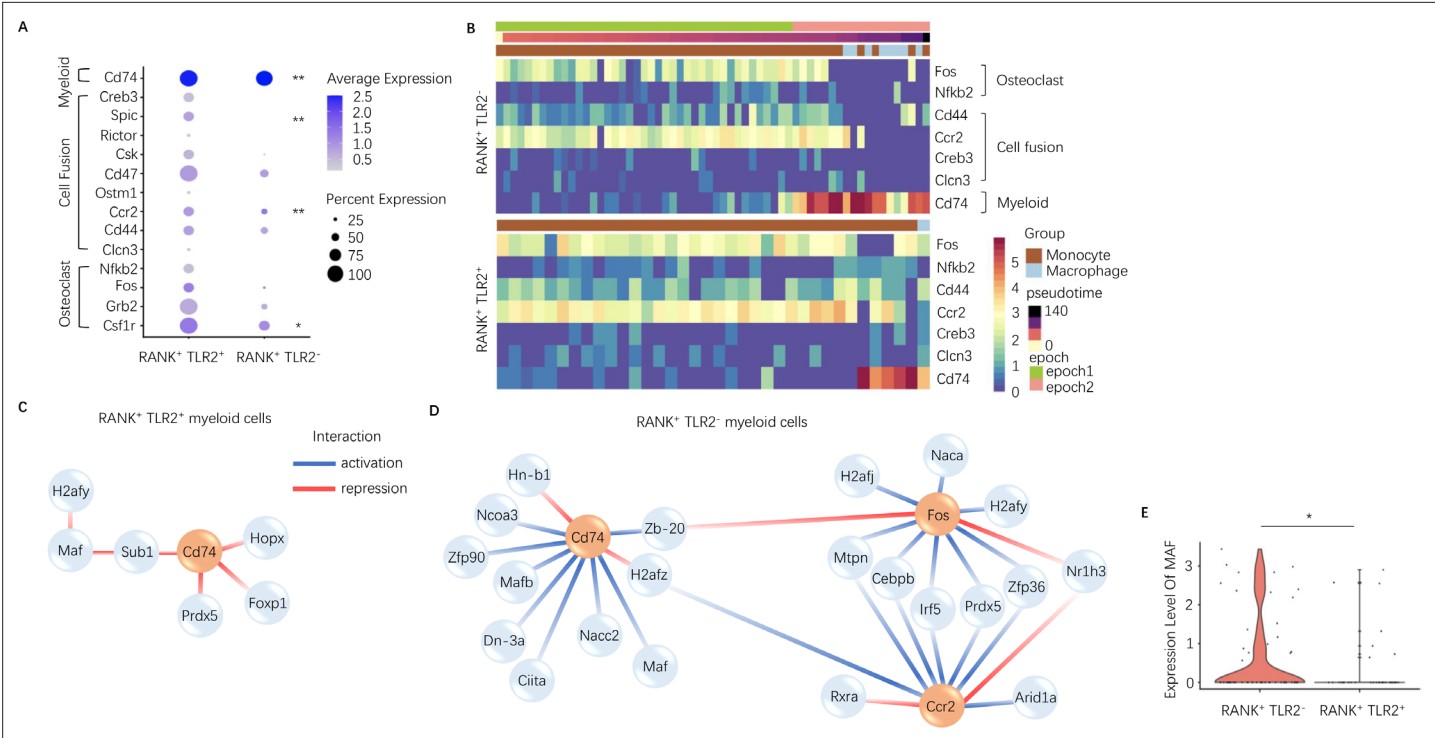

**Figure 2.** Expression of osteoclast fusion genes in RANK⁺TLR2⁺ subset peak during differentiation to a TRAP⁺ linage, but not in the RANK⁺TLR2⁻ subset. (**A**) Dot plot of average expression of select myeloid, cell fusion, and osteoclast differentiation genes in RANK⁺TLR2⁺ myeloid cells (left) and RANK⁺TLR2⁻ myeloid cells (right) from epoch 2. The color of each dot corresponds to average expression and the size corresponds to the percentage of cells expressing the gene of interest (n=3, Poisson test with FDR adjustment). (**B**) Heat maps of dynamic expression of select osteoclast differentiation, cell fusion, and myeloid marker genes in RANK⁺TLR2⁻ myeloid cells (top) and RANK⁺TLR2⁺ myeloid cells (bottom). (**C**) Top downstream regulators for *Cd74* in the epoch 2 stage of RANK⁺TLR2⁺ populations. Blue: activation regulation; Red: repression regulation. (**D**) Top downstream regulators for *Cd74, Fos, and Ccr2* in the epoch 2 stage of RANK⁺TLR2⁻ cells. Zb-20: *Zbtb20*, Dn-3a: *Dnmt3a*, Hn-b1: *Hnrnpa2b1*. (**E**) Violin plots of the expression level of macrophage activating factor *Maf* in RANK⁺TLR2⁻ and RANK⁺TLR2⁺ myeloid cells (n=3, Poisson test with FDR adjustment). *p<0.05, **p<0.01.

The online version of this article includes the following figure supplement(s) for figure 2:

**Figure supplement 1.** The dynamic gene expression of those related to cell fusion and osteoclast differentiation in RANK⁺TLR2⁺ and RANK⁺TLR2⁻ populations.

do not fuse into mature multinucleated osteoclasts but rather induce type H vessel formation during bone coupling (*Xie et al., 2014*). Therefore, RANK⁺TLR2⁻ myeloid cells may represent a precursor cell type that maintains the pool of non-fused periosteal TRAP⁺ mononuclear cells, including those that induce type H vessel formation.

To examine the mechanistic differences between RANK⁺TLR2⁺ and RANK⁺TLR2⁻ myeloid cells, we further characterized the underlying dynamic gene regulatory networks (GRNs) in each subpopulation. Interestingly, GRN analysis demonstrated that expression of the myeloid cell marker gene *Cd74* is activated by transcription factors *Maf* and *Mafb* (among others) in RANK⁺TLR2⁻ myeloid cells but is repressed by *Hopx, Foxp1, Prdx5* and indirectly by *Maf* through *Sub1* in RANK⁺TLR2⁺ myeloid cells (*Figure 2C and D*). Expression of *Maf* was significantly higher in RANK⁺TLR2⁻ myeloid cells as compared to the RANK⁺TLR2⁺ subpopulation (*Figure 2E*), implicating *Maf* as a potential driver of non-fusion cell fate. When we examined the downstream targets of *Maf* and *Mafb* in RANK⁺TLR2⁻ myeloid cells we found that as monocytes differentiate into macrophages in this subpopulation *Maf* expression was associated with the activation of expression of two monocyte marker genes (*Ms4a7* and *Fcgr2b*), a regulator of MMP2 activity and ECM abundance (*Timp2*)(*Arpino et al., 2015*) and *Apoe* (*Figure 2— figure supplement 1E*). Genetic deletion of *Fcgr2b* has been found to enhance pro-inflammatory macrophage activity in the CIA mouse (*Bournazos et al., 2016*) and *Apoe* has been shown to induce an anti-inflammatory phenotype in macrophage (*Baitsch et al., 2011*). These results further suggest that RANK⁺TLR2⁻ myeloid cells represent a novel subpopulation involved in anabolic bone formation

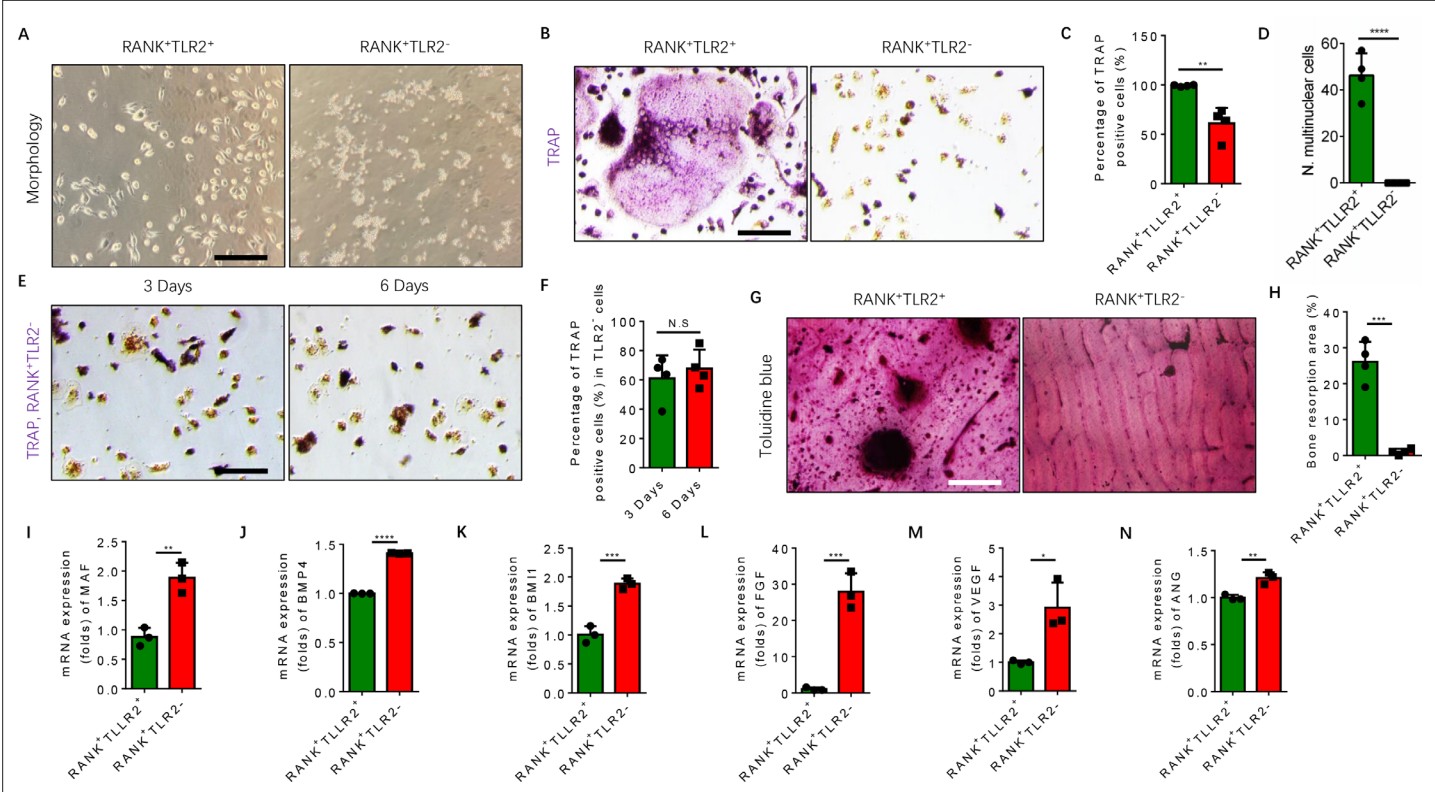

**Figure 3.** RANK+TLR2− myeloid monocytes did not fuse to form osteoclasts after commitment to a tartrate-resistant acid phosphatase-positive (TRAP+) mononuclear lineage. (**A**) Representative images of the morphology of the RANK+TLR2+ and the RANK+TLR2− monocytes after treatment with M-CSF (30 ng/ml) for 3 days (n=3). Scale bar, 0.1 mm. (**B**) Representative images of TRAP staining of RANK+TLR2+ and the RANK+TLR2− monocytes after treatment with M-CSF (30 ng/ml) and RANKL (200 ng/ml) for 3 days. Scale bar, 0.1 mm. (**C**) Quantitative analysis of the percentage of TRAP-positive RANK+TLR2− and RANK+TLR2+ monocytes (n=4, t-test). (**D**) Quantitative analysis of the number of multinuclear cells in the RANK+TLR2+ and the RANK+TLR2− monocytes (n=4, t-test). (**E**) Representative images of TRAP staining of RANK+TLR2− monocytes after treatment with macrophage colony stimulating factor (M-CSF) (30 ng/ml) and RANKL (200 ng/ml) for three days and six days, respectively. Scale bar, 0.1 mm. (**F**) Quantitative analysis of the percentage of TRAP-positive RANK+TLR2− cells. (n=4, t-test). (**G**) Representative images of toluidine blue staining of bone slices after culture with RANK+TLR2+ or RANK+TLR2− monocytes treated with M-CSF (30 ng/ml) and RANKL (200 ng/ml) for 1 week. Scale bar, 0.1 mm. (**H**) The quantitative analysis of bone resorption area in the bone slices (n=4, t-test). (**I–N**) The mRNA expression changes of macrophage activating gene and osteogenesis-related genes, including *Maf, BMP4, BMI1, FGF, VEGF, ANG*, in FACS-sorted RANK+TLR2+ and RANK+TLR2− monocytes (n=3, t-test). All data are means ± SD. N.S=No Significant difference, *p<0.05, **p<0.01, ***p<0.001, ****P<0.0001.

by maintaining the non-fusion status of TRAP+ monocytes, and thus shifting these precursor cells away from an osteoclast fate.

## RANK+TLR2− monocytes differentiate into a TRAP+ lineage, but do not undergo osteoclast fusion

TRAP+ monocytes undergo fusion when attached to the bone surface. Given that our scRNA-seq libraries were prepared from bone marrow cells, TRAP+ cells were attached to the bone and thus were of representation limited in the libraries. To examine the mechanism of TLR2-mediated regulation of osteoclast fusion and differentiation of TRAP+ mononuclear cells, we isolated primary bone marrow monocytes and sorted them for RANK+TLR2+ and RANK+TLR2− subpopulations. We then cultured the two subpopulations and treated each with M-CSF for three days to commit them to a TRAP+ lineage. After such treatment we found that the morphology of RANK+TLR2+ monocytes was markedly different from that of RANK+TLR2− cells (*Figure 3A*). Importantly, when RANKL was added with M-CSF for three days, cell fusion leading to TRAP+ osteoclast formation was observed in RANK+TLR2+ cells, but not in RANK+TLR2− monocytes (*Figure 3B–D*). The RANK+TLR2− monocytes were cultured for six days with M-CSF and RANKL, but still did not undergo cell fusion (*Figure 3E and F*). Moreover, toluidine blue staining demonstrated that the bone resorption areas on bone slices co-cultured with

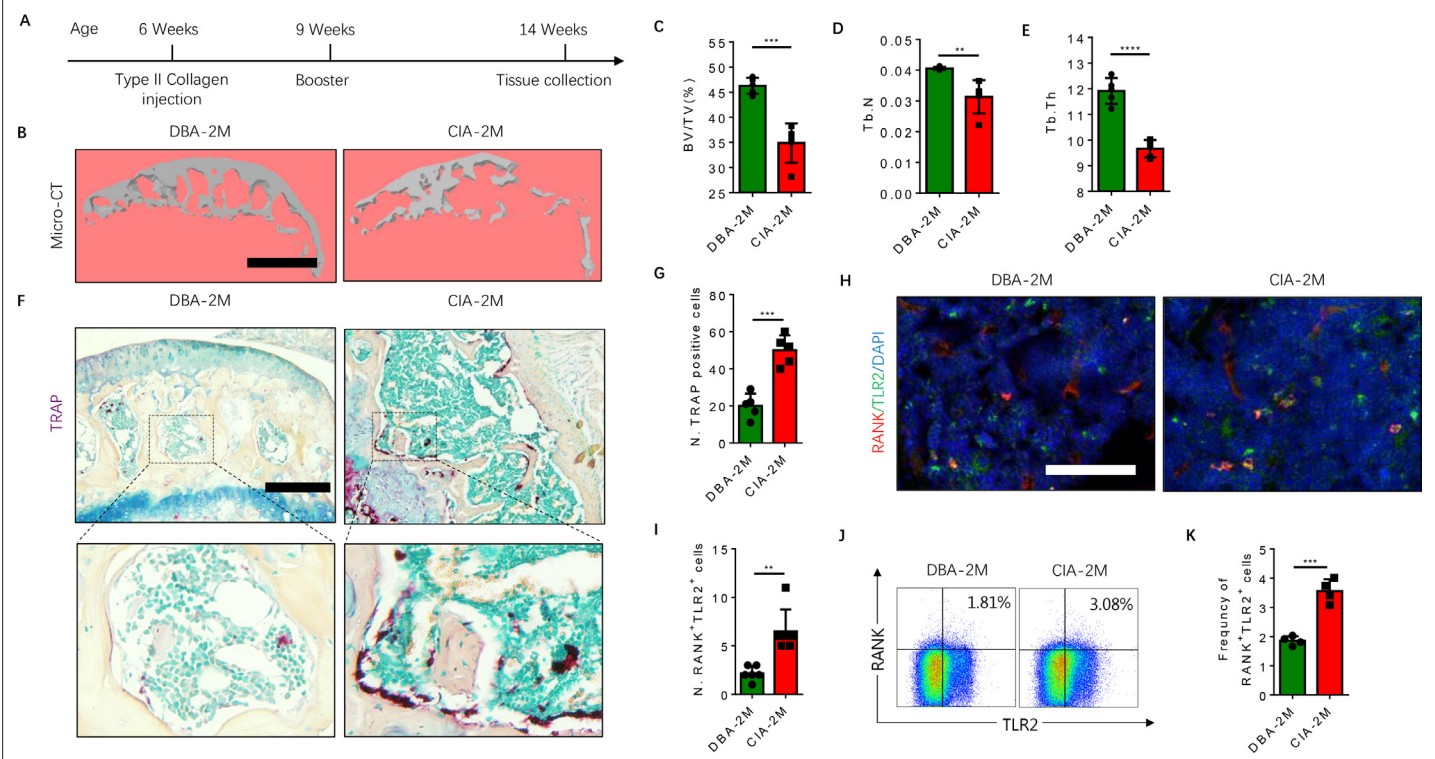

**Figure 4.** The frequency of RANK⁺TLR2⁺ myeloid monocytes is greater in CIA mice. (**A**) A schematic diagram illustrating the experimental regimen for arthritis induction in the CIA model of RA. (**B**) Representative 3D µCT images of the subchondral bone of the medial compartment of the tibia (sagittal view) of DBA (control) and CIA mice after 2 months of immunization (DBA-2M and CIA-2M, respectively). Scale bar, 0.1 mm. (**C–E**) Quantitative analysis of bone volume/tissue volume (BV/TV), trabecular number (Tb. N), trabecular thickness (Tb. Th) of the CIA-2M and DBA-2M mice (n=5, t-test). (**F**) Representative images of TRAP staining in knee joints. Scale bar, 0.5 mm. (**G**) Quantitative analysis of the number of tartrate-resistant acid phosphatase-positive (TRAP⁺) cells (n=5, t-test). (**H**) Representative images of co-immunofluorescence staining for RANK (red), TLR2 (green), and DAPI (blue) staining of nuclei in knee joints. Scale bar, 0.5 mm. (**I**) Quantitative analysis of the number of double-positive RANK and TLR2 (n=6, t-test). (**J**) Representative flow cytometry plots with the percentage of RANK⁺TLR2⁺ cells in bone marrow. (**K**) Frequency of RANK⁺TLR2⁺ cells in the bone marrow (n=4, t-test). All data are means ± SD. **p<0.01, ***p<0.001, ****p<0.0001.

the RANK⁺TLR2⁺ subpopulation were significantly larger than those co-cultured with the RANK⁺TLR2⁻ subpopulation in a resorption pit assay (*Figure 3G and H*).

*Maf* is the key transcription factor for macrophages, and its expression is approximately twofold higher in the RANK⁺TLR2⁻ subpopulation compared to the RANK⁺TLR2⁺ subpopulation (*Figure 3I*), suggesting the former cells are macrophages in nature, while the latter cells are more competent for fusion and osteoclast formation. TRAP⁺ mononuclear cells secrete PDGF-BB to induce type H vessel formation for anabolic bone formation (*Kusumbe and Adams, 2014*). Indeed, the expression of bone formation factor BMP4 and BMI1 and angiogenesis factor FGF, VEGF, and angiogenin were significantly greater in the RANK⁺TLR2⁻ subpopulation relative to the RANK⁺TLR2⁺ subpopulation (*Figure 3J–N*). These results suggest that expression of *Maf* in the absence of TLR2 in RANK⁺ monocytes retains these cells as TRAP⁺ mononuclear cells that promote angiogenesis and bone anabolism, rather than promoting cell fusion and osteoclast maturation and thus bone catabolism.

## A RANK⁺TLR2⁺ monocyte subpopulation is increased in association with osteoclastic resorption of subchondral bone in an RA mouse model

To further examine that the elevated RANK⁺TLR2⁺ subpopulation converted autoimmune to osteoclast destruction of joints, we generated the well-regarded collagen-induced arthritis (CIA) model by injecting an emulsion of type II collagen (2 mg/ml) into the tail subcutaneous tissue, along with a booster injection 21 days later, as previously described (*Brand et al., 2007*). The knee joints were

harvested from the CIA mice for analysis 2 months after the initial immunization (*Figure 4A*). We then analyzed the subchondral bone of the joint by µCT, and we found that the bone volume/tissue volume (BV/TV), trabecular bone number (Th. N) and thickness (Tb. Th) were significantly lower in the CIA mice relative to the vehicle-injection control mice (*Figure 4B–E*). Histological staining of knee joint sections showed the population of TRAP+ osteoclasts was significantly greater in the subchondral bone of the CIA mice relative to control mice (*Figure 4F and G*).

As we previously showed that sialylation of TLR2 induces osteoclast fusion (*Dou et al., 2022*), we next examined whether TLR2 expression is associated with a greater number of osteoclasts in the CIA mice versus the controls. By co-immunostaining we found that the co-localization of TLR2 with RANK was significantly greater in CIA mice relative to vehicle-treated mice (*Figure 4H*), suggesting expression of TLR2 was increased in RANK+ monocytes during arthritis progression. To confirm this observation, we isolated bone marrow and performed flow cytometry analysis, and we found that the frequency of the RANK+TLR2+ population was significantly greater in the CIA mice relative to the vehicle-injected controls (*Figure 4J and K*). Taken together, the greater frequency of the RANK+TLR2+ monocyte subpopulation and the greater number of TRAP+ cells in the CIA mice versus the controls indicate a critical role of TLR2 in osteoclast fusion and subsequent joint destruction.

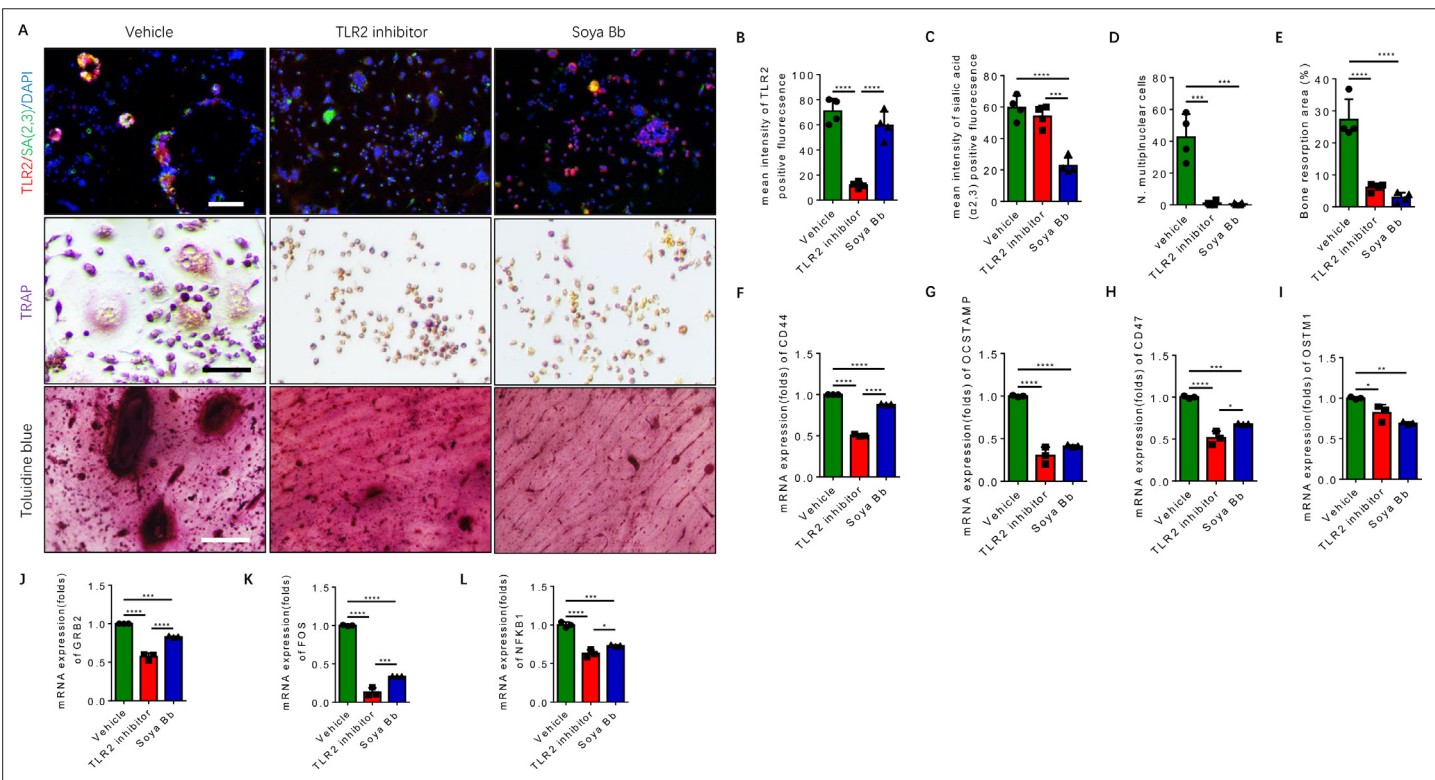

**Figure 5.** Inhibition of TLR2 activity or its sialylation blocks the osteoclastogenesis of RANK+TLR2+ monocytes. (**A**) Upper raw: Representative images of the immunofluorescence co-staining of TLR2 (red) and α(2,3) sialylation (green) of RANK+TLR2+ monocytes treated with vehicle (PBS), a TLR2 inhibitor (C29, 150 µM), or an α(2,3) sialyltransferase inhibitor (soyasaponin Bb, 10 mg/kg), respectively, for three days. DAPI (blue) for the nuclei. Middle raw: Representative images of TRAP staining of RANK+TLR2+ monocytes treated with PBS, C29, or soyasaponin Bb, respectively, for 3 days. Bottom raw: Representative images of toluidine blue staining of bone slices cultured with RANK+TLR2+ monocytes for 1 week, that had been treated with PBS, C29, or soyasaponin Bb, respectively, for 1 week prior. Scale bar, 0.1 mm. (**B–C**) Quantitative results of the mean intensity of TLR2− and α(2,3) sialylation-positive fluorescence in the RANK+TLR2+ monocytes treated with PBS, C29, or soyasaponin Bb, respectively (n=4, one-way ANOVA with Tukey's multiple comparisons test). (**D**) Quantitative analysis for multinuclear cells among the RANK+TLR2+ cells treated with PBS, C29, or soyasaponin Bb, respectively (n=4, one-way ANOVA with Tukey's multiple comparisons test). (**E**) Quantitative analysis of the bone resorption area of bone slices cultured with RANK+TLR2+ cells treated with PBS, C29, or soyasaponin Bb, respectively (n=4, one-way ANOVA with Tukey's multiple comparisons test). (**F–I**) The mRNA expression of the cell fusion-related genes *Cd44, Ocstamp, Cd47,* and *Ostm1* in RANK+TLR2+ monocytes treated with PBS, C29, or soyasaponin Bb, respectively (n=3, one-way ANOVA with Tukey's multiple comparisons test). (**J–L**) The mRNA expression of the osteoclast differentiation-related genes *Grb2, Fos,* and *Nfkb1* in RANK+TLR2+ monocytes treated with PBS, C29, or soyasaponin Bb, respectively (n=3, one-way ANOVA with Tukey's multiple comparisons test). All data are means ± SD. *p<0.05, **p<0.01, ***p<0.001, ****p<0.0001.

## Inhibition of TLR2 sialylation blocks osteoclast fusion in RANK+TLR2+ monocytes

To characterize the downstream function of TLR2 and of sialylation in RANK+TLR2+ monocytes based on our previous finding (*Dou et al., 2022*), the RANK+TLR2+ cells were treated with the TLR2 inhibitor C29 and the sialyltransferase inhibitor soyasaponin Bb (*Hsu et al., 2005*; *Wu et al., 2001*). Co-immunofluorescence staining showed that the expression of TLR2 significantly decreased with C29 treatment relative to vehicle control, whereas sialylation levels decreased significantly upon soyasaponin Bb treatment while TLR2 expression was unchanged (*Figure 5A–C*). TRAP staining demonstrated that no multinuclear osteoclasts were formed when the cells were treated with either C29 or soyasaponin Bb, while a significant number of multinuclear osteoclasts were formed with vehicle treatment (*Figure 5A and D*). In the bone resorption pit assay, toluidine blue staining further showed that the resorbed areas in the C29- or soyasaponin Bb-treated groups were significantly decreased relative to the vehicle-treated control group (*Figure 5A and E*).

Based on the analysis of our scRNA-seq data (*Figure 2—figure supplement 1A-D*), the top representative genes of cell fusion and osteoclast differentiation were chosen to further examine the functional activity of TLR2 sialylation. The expression of cell fusion genes, including *Cd44, Ocstamp, Cd47,* and *Ostm1*, were significantly lower in RANK+TLR2+ monocytes treated with C29 or soyasaponin Bb relative to the vehicle group (*Figure 5F–I*). Likewise, the expression of osteoclast differentiation genes, including *Grb2, Fos,* and *Nfkb1*, were also significantly lower in RANK+TLR2+ monocytes treated with C29 or soyasaponin Bb relative to the vehicle-treated group (*Figure 5J–L*). Taken together, these data show that blocking TLR2 signaling, either by inhibiting its activity or its sialylation, inhibits the fusion and osteoclastogenesis of RANK+TLR2+ monocytes, thus reducing bone resorption.

## A significant increase of sialylation by specific sialyltransferases occurs in CIA mice

To investigate the role of sialylation of TLR2 in RA-associated joint destruction, we measured the levels of sialic acid α(2,3) of bone marrow cells, and found the levels were significantly higher in the CIA mouse model relative to vehicle-injected control mice (*Figure 6A and B*). Immunofluorescence staining of knee joint sections also showed that the fluorescence intensity of sialylation was significantly greater in the subchondral bone of CIA mice relative to control mice (*Figure 6C and D*). In addition, the expression of c-Fos, a transcription factor critical for the regulation of osteoclast differentiation (*Park et al., 2017*), was absent in the RANK+TLR2⁻ late monocyte and macrophage but continued to be expressed throughout the differentiation trajectory of the RANK+TLR2+ myeloid subpopulation (*Figure 6E*). Immunostaining and qPCR demonstrated that c-Fos expression was significantly greater in CIA mice relative to control mice (*Figure 6F and H*), and the expression level in the RANK+TLR2+ subpopulation was significantly higher relative to the RANK+TLR2⁻ subpopulation (*Figure 6I*). This greater expression of c-Fos in the RANK+TLR2+ subpopulation suggests their contribution to the pool of cells involved in osteoclast fusion and differentiation.

To identify the specific sialyltransferase(s) that mediates the increased sialylation, we analyzed the dynamical expression of all sialyltransferases in both RANK+TLR2+ and RANK+TLR2⁻ monocytes in our single-cell libraries. We found that sialyltransferases *St3gal1, St3gal4,* and *St6gal1* were expressed in both subpopulations (*Figure 6E*). Co-immunostaining of ST3GAL4 with RANK demonstrated that RANK+ST3GAL4+ cells in the subchondral bone significantly increased in CIA mice relative to control mice (*Figure 6J and K*). However, the co-immunostaining did not show that ST3GAL1 was colocalized with RANK-positive myeloid cells (*Figure 6—figure supplement 1A–C*), suggesting ST3GAL1 was uninvolved. We, therefore, further characterized the transcriptional regulation of ST3GAL4 by c-Fos, as ST6GAL1 does not mediate the sialylation modification of sialic acid α(2,3). By CHIP assay we confirmed that c-Fos activates the transcription of ST3GAL4 by binding to two of the putative c-Fos binding sites, which was predicted by using the JASPAR database (*Castro-Mondragon et al., 2022*; *Figure 6L and M*). We found that there are three putative c-Fos binding sites in the *St3gal4* promoter.

## Inhibition of sialylation reduces the RANK+TLR2+ subpopulation and protects CIA mice from joint destruction

To test whether inhibition of sialylation could reduce the pool of RANK+TLR2+ monocytes and thus joint destruction in vivo, CIA mice were prophylactically treated with orally administered soyasaponin

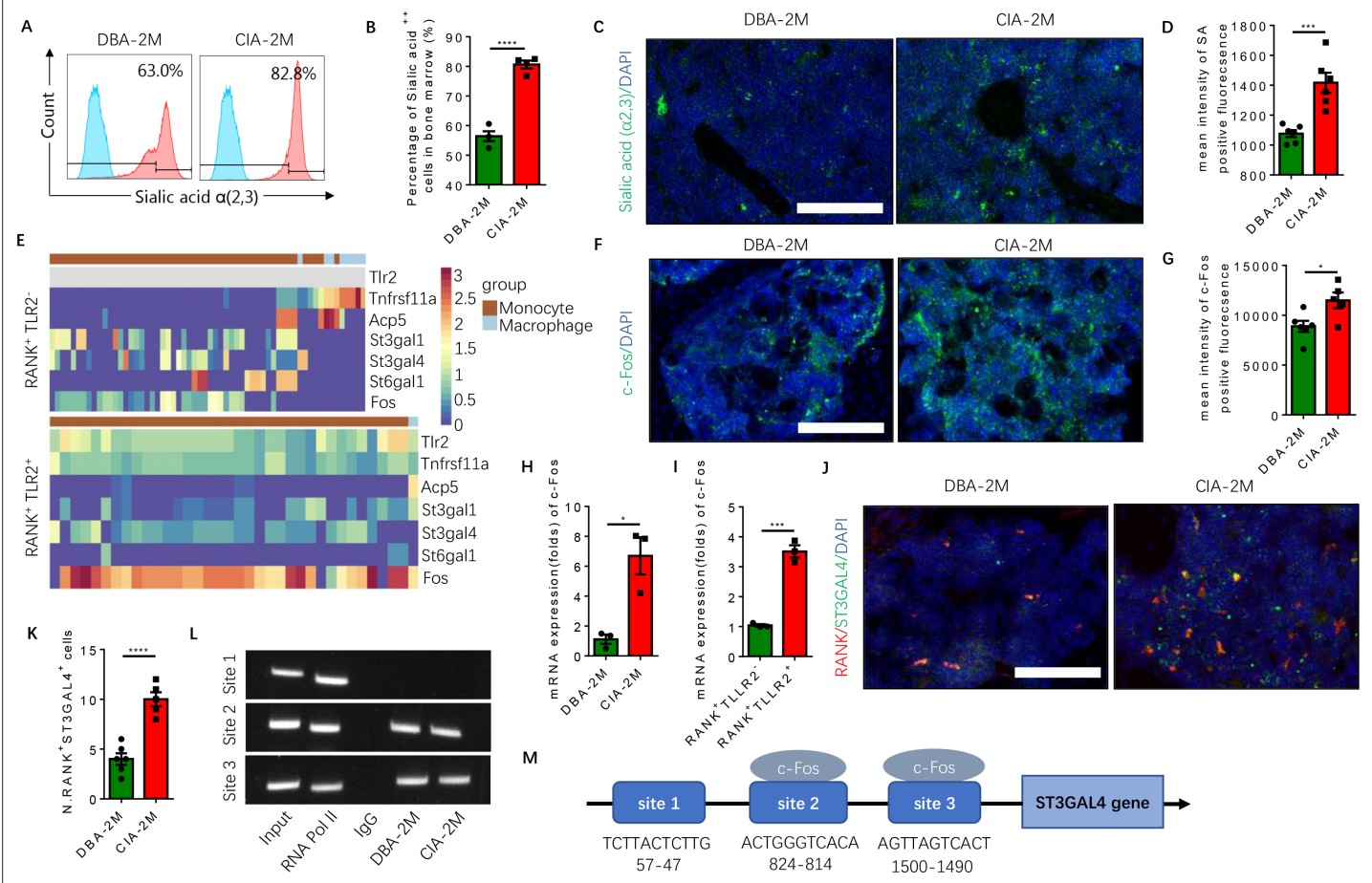

**Figure 6.** Elevated α(2,3) sialylation in RANK⁺TLR2⁺ monocytes is associated with an increase in the transcription of sialyltransferases in CIA mice.
(**A**) Representative histograms of sialic acid α(2,3)-modified cells in the bone marrow of DBA control mice and CIA mice 2 months after the first immunization. (**B**) Quantitative analysis of the frequency of sialic acid α(2,3)-positive cells in DBA mice and CIA mice (n=4, *t*-test). (**C**) Representative images of the biotin fluorescence staining for sialic acid α(2,3) (green) and of DAPI staining (blue) of the knee joint sections from DBA mice and CIA mice 2 months after the first immunization. Scale bar, 0.5 mm. (**D**) Quantitative analysis of the mean intensity of sialic acid α(2,3)-positive fluorescence of the images represented in (**C**) (n=6, *t*-test). (**E**) Heat maps of dynamic expression of marker genes *Tlr2, Tnfrsf11a, Acp5*, sialyltransferases, and *Fos* in the RANK⁺TLR2⁻ myeloid cells (top) and RANK⁺TLR2⁺ myeloid cells (bottom). (**F**) Representative images of the immunofluorescence staining for c-Fos (green) of the knee joint section from DBA mice and CIA mice 2 months after the first immunization. Scale bar, 0.5 mm. (**G**) Quantitative analysis of the mean intensity of c-Fos-positive fluorescence in (**F**) (n=5, *t*-test). (**H–I**) The mRNA expression changes of c-Fos in the knee joint tissue of CIA and DBA mice (**H**) or in the RANK⁺TLR2⁻ and RANK⁺TLR2⁺ monocytes (**I**) (n=3, *t*-test). (**J**) Representative images of the immunofluorescence co-staining for RANK (red) and ST3GAL4 (green) of the knee joint sections from DBA mice and CIA mice 2 months after the first immunization. (**K**) Quantitative analysis of the RANK⁺ST3GAL4⁺ cells in (**J**) (n=5, *t*-test). (**L**) The electrophoresis of CHIP assay for the binding sites of c-Fos in the promoter sequence of the *St3gal4* gene. (**M**) The diagram of different locations and sequences of the putative binding sites for c-Fos in the promoter of *St3gal4* gene. *p<0.05, ***p<0.001, ****p<0.0001.

The online version of this article includes the following source data and figure supplement(s) for figure 6:

**Source data 1.** Gel blot of CHIP assay of St3gal4 gene expression.

**Figure supplement 1.** Different expression of ST3GAL1 in RANK positive population.

Bb once a day for 2 months, beginning from the first day of type II collagen immunization (*Figure 7A*). Flow cytometry analysis showed that the percentage of α(2,3) sialylation positive cells were significantly lower in the soyasaponin Bb treated CIA mice relative to vehicle-treated controls (*Figure 7B and C*). Further, the BV/TV of the knee joint subchondral bone and the trabecular bone number was significantly greater in the soyasaponin Bb-treated group versus the vehicle-treated mice (*Figure 7D–F*). Importantly, the number of TRAP⁺ osteoclastic cells was significantly lower with soyasaponin Bb treatment relative to vehicle treatment (*Figure 7D and G*).

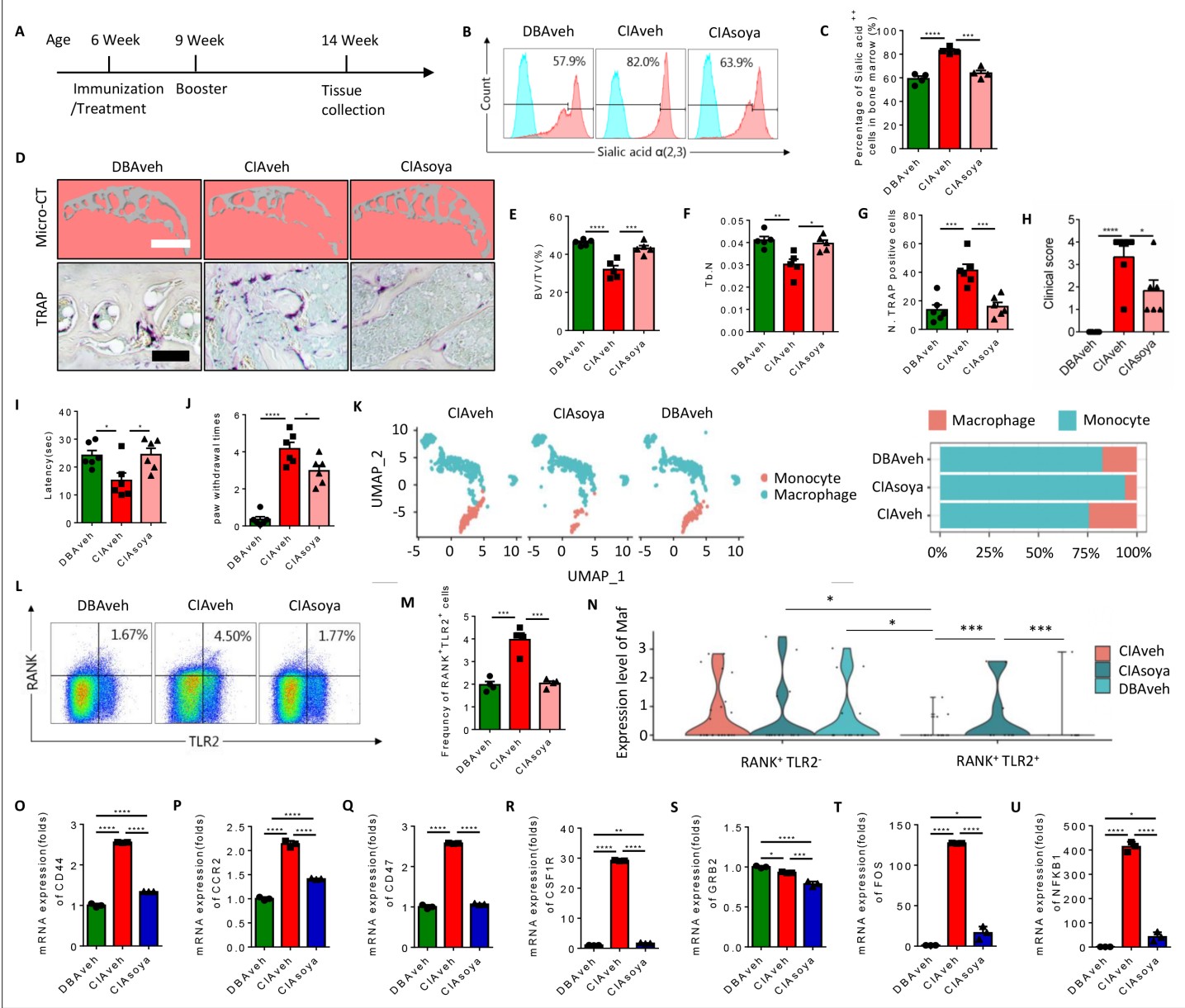

**Figure 7.** Inhibition of α(2,3) sialylation blocks osteoclast fusion and joint destruction in CIA mice. (**A**) A schematic diagram illustrating the time frames for immunization and soyasaponin Bb or vehicle (PBS) treatment, booster injection, and analysis. (**B**) Representative histograms for sialic acid α(2,3) modified cells in the bone marrow of CIA mice treated with PBS or soyasaponin Bb relative to DBA mice. (**C**) The frequency of sialic acid α(2,3) positive cells in CIA mice treated with PBS or soyasaponin Bb relative to DBA mice (n=4, one-way ANOVA with Tukey's multiple comparisons test). (**D**) Upper raw: The representative 3D reconstruction images for subchondral bone of knee joints of DBA mice and CIA mice treated with PBS or soyasaponin Bb for 2 months. Lower raw: Representative images of the TRAP staining for the knee joint sections in DBA mice and CIA mice treated with PBS or soyasaponin Bb. Scale bar, 0.1 mm. (**E–F**) Quantitative analysis of bone volume/tissue volume (BV/TV) (**E**) or Tb. N (**F**) in μCT scan of CIA mice treated with PBS or soyasaponin Bb relative to DBA mice (n=5, one-way ANOVA with Tukey's multiple comparisons test). (**G**) Quantitative analysis of TRAP-positive cells in the knee joint of DBA mice and CIA mice treated with PBS or soyasaponin Bb (n=6, one-way ANOVA with Tukey's multiple comparisons test). (**H**) The clinical score of the inflammation paws in CIA mice treated with PBS or soyasaponin Bb relative to DBA mice (n=6, one-way ANOVA with Tukey's multiple comparisons test). (**I**) The latency time for withdrawal of the paw after light beam-induced heat stimulation in CIA mice treated with PBS or soyasaponin Bb relative to DBA mice (n=6, one-way ANOVA with Tukey's multiple comparisons test). (**J**) The frequency of paw withdrawals after the stimulation of the von Frey hair (0.4 g) in CIA mice treated with PBS or soyasaponin Bb relative to DBA mice (n=6, one-way ANOVA with Tukey's multiple comparisons test). (**K**) UMAP embedding of cell type annotation of RANK+ monocytes for each sample group. Bar plot of relative cell type abundance of RANK+ monocytes in each sample group. (**L**) Representative flow cytometry plots for RANK+TLR2+ cells in bone marrow in CIA mice treated with PBS or soyasaponin Bb relative to DBA mice. (**M**) The frequency of RANK+TLR2+ cells in bone marrow in CIA mice treated with PBS or soyasaponin Bb relative to DBA mice (n=4, one-way ANOVA with Tukey's multiple comparisons test). (**N**) Violin plot of *Maf* expression in RANK+ myeloid cells in the TLR2− and

*Figure 7 continued on next page*

*Figure 7 continued*

TLR2$^+$ subsets across sample groups (n=3, Poisson test with FDR adjustment). (**O–Q**) The fold mRNA expression of the cell fusion-related genes *Cd44*, *Ccr2*, *Cd47* in CIA mice treated with PBS or soyasaponin Bb relative to DBA mice (n=3, one-way ANOVA with Tukey's multiple comparisons test). (**R–U**) The mRNA expression changes of the osteoclast differentiation-related genes *Csf1r*, *Grb2*, *Fos*, *Nfkb1* in CIA mice treated with PBS or soyasaponin Bb relative to DBA mice (n=3, one-way ANOVA with Tukey's multiple comparisons test). *p<0.05, **p<0.01, ***p<0.001, ****p<0.0001.

The online version of this article includes the following figure supplement(s) for figure 7:

**Figure supplement 1.** Clustering of single-cell RNA-sequence (scRNA-seq) data in groups.

In terms of outcome, the qualitative scoring of the severity of paw inflammation showed that the swelling and thickness of the paw were significantly lower in the soyasaponin Bb-treated group compared to the vehicle-treated mice (*Figure 7H*). And by the Hargreaves and the von Frey pain behavior tests, we found that the latency was elevated, and the paw withdraw times were rescued, in the CIA mice treated with soyasaponin Bb compared to vehicle mice (*Figure 7I and J*).

We further analyzed our single-cell sequencing data libraries to validate the changes of RANK$^+$ subsets upon soyasaponin Bb treatment (*Figure 7—figure supplement 1A*), and we found that the percentage of monocytes was greater, while the transition to macrophage cells was lower, in the CIA mice treated with soyasaponin Bb mice relative to the vehicle treated mice (6.17% relative abundance of macrophages in the CIA treated group as compared to 24.3% in the untreated RA group) (*Figure 7K*). Moreover, by flow cytometry analysis we found that the size of the RANK$^+$TLR2$^+$ subset was significantly greater in CIA mice with vehicle treatment relative to healthy control mice, but the larger pool size was effectively blunted by soyasaponin Bb treatment (*Figure 7L and M*). As shown above, *Maf* expression is significantly higher in the RANK$^+$TLR2$^-$ subset than the RANK$^+$TLR2$^+$ subset, but in the soyasaponin Bb treatment group we observed *Maf* expression in the RANK$^+$TLR2$^+$ subset which is significantly greater than *Maf* expression in the RANK$^+$TLR2$^+$ control or RA groups (*Figure 7N*), suggesting such expression maintains the RANK$^+$TLR2$^+$ subset in the myeloid lineage and thus reduces their osteoclast differentiation. We further examined the expression of the top representative genes of cell fusion and osteoclast differentiation identified in our single-cell sequencing data (*Figure 2—figure supplement 1A-D*). We found that soyasaponin Bb treatment was associated with significantly lower expression of both the cell fusion-related genes (*Cd44, Ccr2, Cd47*) (*Figure 7O–Q*) and the osteoclast differentiation genes (*Csf1r, Grb2, Fos, Nfkb1*) in the RANK$^+$TLR2$^+$ subset compared to the same cells in the vehicle treat CIA mice (*Figure 7R–U*). Taken together, our results show that pharmacological inhibition of α(2,3) sialylation in CIA mice results in less osteoclast fusion and differentiation of the RANK$^+$TLR2$^+$ subset and thus less joint destruction.

## Discussion

The current conceptual model in the field is that under the induction of RANKL signaling RANK$^+$ myeloid cells commit to a TRAP$^+$ lineage and subsequently undergo cell-cell fusion during osteoclast formation (*Boyle et al., 2003*). And previously we found that sialylation of TLR2 allows it to be a receptor for Siglec15, thus promoting RANKL-induced osteoclast formation (*Dou et al., 2022*). But importantly we found here that even though both RANK$^+$TLR2$^+$ or RANK$^+$TLR2$^-$ myeloid monocytes can commit to a TRAP$^+$ lineage downstream of RANKL signaling, only the RANK$^+$TLR2$^+$ subset is competent to undergo cell-cell fusion during osteoclast formation. However, when α(2,3) sialylation is pharmacologically inhibited in vitro or in vivo, the RANK$^+$TLR2$^+$ subset can differentiate into a TRAP$^+$ lineage and will not fuse.

These findings provide possible answers for some longstanding questions in bone biology while providing a potential therapeutic target for the treatment of joint destruction in RA. Notably, TRAP$^+$ cells at the periosteal surface have been observed to maintain their mononuclear cell type during the regulation of periosteal bone formation (*Gao et al., 2019*). Our findings suggest that these periosteal TRAP$^+$ mononuclear cells could be derived either from a RANK$^+$TLR2$^-$ myeloid subset or from a RANK$^+$TLR2$^+$ subset that lacks α(2,3) sialylation. Also, as sufficient estrogen levels are essential to maintain the lifespan balance between preosteoclasts and mature osteoclasts, during menopause preosteoclasts rapidly fuse and osteoclast formation is markedly increased. As the incidence of RA markedly increases in post-menopausal women (*Carlsten, 2005*; *Sapir-Koren and Livshits, 2017*), an increase of the RANK$^+$TLR2$^+$ subset and of α(2,3) sialylation could explain this increased incidence,

while also contributing to both joint destruction in RA and osteoporosis in such women. Future studies are needed to explore the potential role of estrogen signaling in the regulation of the size of the RANK$^+$TLR2$^+$ subset and of α(2,3) sialylation.

In this current study, we found that the size of the RANK$^+$TLR2$^+$ subset was greater in the CIA mouse model of RA and that prophylactic treatment with the α(2,3) sialylation inhibitor soyasaponin Bb resulted in significantly less osteoclast formation and joint destruction compared to vehicle-treated CIA mice. Interestingly, while the treatment with soyasaponin Bb resulted in a smaller size of the RANK$^+$TLR2$^+$ subset, suggesting that inhibition of α(2,3) sialylation also regulates the size of the RANK$^+$TLR2$^+$ subsets. These findings complement those that have been previously reported that showed that treatment with agonists for TLR2 increases osteoclast formation with significant bone loss (*Souza and Lerner, 2019*; *Elshabrawy et al., 2017*), and that TRAP$^+$ preosteoclasts derived from TLR2 knockout mice do not differentiate into mature osteoclasts upon RANKL stimulation (*Nishimura et al., 2016*). Thus, together these results show that TLR2 is essential for the fusion of RANK$^+$ myeloid cells during osteoclast formation, and that sialylation of TLR2 initiates the cell-cell recognition that occurs in osteoclast fusion. Importantly, inhibition of α(2,3) sialylation could be a potential target for RA-associated joint destruction, though further studies are needed to explore if the therapeutic use of such inhibitors can slow or even reverse the progression of such joint damage.

RANK$^+$ myeloid monocytes, regardless of their TLR2 expression, can differentiate into TRAP$^+$ osteoclastic lineage cells. Therefore, there are actually three different subtypes of TRAP$^+$ cells - RANK$^+$TLR2$^-$, RANK$^+$TLR2$^{+un}$(unsialylated), and RANK$^+$TLR2$^{+sa}$(sialylated) - and they possess different roles in bone homeostasis. Characterization of the specific function of each subtype and their relationship with osteoblast differentiation, sensory innervation and angiogenesis will provide fundamental insights into bone biology. Notably, RANK$^+$TLR2$^-$ myeloid monocytes are a novel subset identified in this study. Analysis of our scRNA-seq libraries showed that *Maf* is highly expressed in these cells, and some of the important osteoclast fusion and differentiation genes are not expressed, which partly explains their non-fusion nature and their potential function. Importantly, these cells express bone anabolism-related genes, suggesting that although this novel subset can be committed to a TRAP$^+$ lineage they are not preosteoclasts and that they may function in the communication with osteoblastic cells for their osteogenesis during bone coupling. The TRAP$^+$ mononuclear cells from either the RANK$^+$TLR2$^{+un}$ subset or the RANK$^+$TLR2$^{+sa}$ subset are preosteoclasts. The α(2,3) sialylation controls the balance between preosteoclasts and osteoclasts in coupling osteoclastic bone resorption with type H vessel formation and osteogenesis (*Xie et al., 2014*). There are two α(2,3) sialyltransferases, *St3gal1* and *St3gal4*, expressed in RANK$^+$ myeloid monocytes during their differentiation to a TRAP$^+$ lineage. The balance between preosteoclasts and osteoclasts is likely regulated by the expression of these two α(2,3) sialyltransferases (*Hsu et al., 2005*; *Wu et al., 2001*) or sialidase, though cell-type specific knockout experiments are required to validate that notion. Even so, the regulation of the sizes of different RANK$^+$ subsets maintains the balance between osteoclast bone resorption and osteoblast bone formation, which is particularly important for joints. Leveraging this insight may allow for safer and more effective therapeutics to prevent or slow down joint destruction during RA progression, while potentially expanding such therapies to other forms of bone loss, such as age-related osteoporosis.

## Materials and methods
### Mice and in vivo treatment
DBA/1 J mice were purchased from the Jackson Laboratory (strain number: 000670). The CIA model was established by injecting chicken type II collagen subcutaneously into the mouse tail of female DBA/1 J mice at the 6 week of age. In brief, we prepared the collagen emulsion by homogenizing a chicken type II collagen solution (Chondrex) with Complete Freund's Adjuvant (Chondrex) at high speed (15,000 rpm) following the manufacturer's instructions. We injected 100 µg of the collagen emulsion into the tail subcutaneously, followed by a booster injection (100 µg collagen emulsified with Incomplete Freund's Adjuvant [Chondrex]) at day 21 post-immunization. For the prophylactic treatment studies, the CIA mice were administrated with either vehicle (PBS) or soyasaponin Bb (10 mg/kg) intragastrically once per day for 2 months from the first day of collagen immunization. The mice were killed for further analysis at 2 months after the first immunization by using an overdose of isoflurane.

C57BL/6 J (wild-type, WT) mice were purchased from the Jackson Laboratory (strain number: 000664). Two-month-old WT mice were killed by using an overdose of isoflurane. The femur and tibia bone were isolated for primary monocyte collection and further experiments.

All animals were kept in the Animal Facility of the Johns Hopkins University School of Medicine. The animal protocol was approved by the Institutional Animal Care and Use Committee of Johns Hopkins University, Baltimore, MD, USA (MO21M276). The ARRIVE guideline has been followed.

## µCT

The joint tissues were collected from mice upon their killing and then fixed with 10% buffered formalin phosphate for 48 hr and transferred into PBS. Then the samples were scanned by high-resolution µCT (Skyscan1172). The parameters were set at a voltage of 55 kVp, a current of 163 µA, and a resolution of 6.0 µM per pixel. The software NRecon v1.6 and CTAn v1.9 (Skyscan) were used to reconstruct and analyze the data. Sagittal images of knee joints were used to perform three-dimensional histomorphometric analyses of subchondral bone. Three-dimensional structural parameters analyzed included BV/TV (trabecular bone volume per tissue volume), Tb. Th (trabecular bone thickness), Tb. N (trabecular bone number). Five consecutive sagittal-oriented images were used to show the three-dimensional reconstruction of the subchondral bone of knee joints using three-dimensional model visualization software, CTVol v2.0 (Skyscan).

## Immunohistochemistry and histomorphometry

The knee joint tissues of the mice were collected after upon killing and fixed in 10% buffered formalin phosphate for 48 hr. Samples were then decalcified in 10% ethylenediaminetetraacetic acid (EDTA, pH 7.4) for 2 weeks and embedded in paraffin or optimal cutting temperature (OCT) compound (Sakura Finetek). Four µm thick, sagittal-oriented sections of paraffin-embedded knee joints were processed for Tartrate-resistant acid phosphatase (TRAP) staining using a standard protocol (Sigma-Aldrich). Ten µm thick, sagittal-oriented sections of OCT-embedded knee joints were processed for immunofluorescence staining. The sections were incubated with primary antibodies against RANK (1:100, ab13918, Abcam), TLR2 (1:100, mAb12276, Cell Signaling), Maackia Amurensis Lectin (1:100, B-1265–1, Vector Laboratories), ST3GAL4 (1:100, 13546–1-AP, Proteintech), c-Fos (1:100, mAb2250, Cell Signaling), or ST3GAL1 (1:50, PA5-21721, Thermo Fisher Scientific) overnight at 4 °C. Then, the samples were incubated with secondary antibodies and DAPI (1:250, H-1200, Vector) in the dark for 1 hr at room temperature. The fluorescence images were taken by fluorescence microscopy (Olympus BX51, DP71) or confocal microscopy (Zeiss LSM 880) and analyzed by ImageJ software (National Institutes of Health, Bethesda).

## Osteoclast differentiation assay

Mouse primary bone marrow monocytes (BMMs) were isolated from femur and tibia bone marrow upon the killing of the mice. We stimulated the BMMs with M-CSF (30 ng/ml, Amizona Scientific LLC, AM10003-600) and RANKL (200 ng/ml, Amizona Scientific LLC, AM10003-500) for osteoclast differentiation. For TRAP staining, BMMs were incubated in 24-well plates at a density of $5 \times 10^4$ cells per well with α-MEM (Gibco, Thermo Fisher Scientific) supplemented with 10% FBS (Foetal Bovine Serum, Gibco, Thermo Fisher Scientific) and 1% penicillin-streptomycin (Gibco, Thermo Fisher Scientific), as well as M-CSF (30 ng/ml) and RANKL (200 ng/ml). Cells were incubated at 37°C in a 5% $CO_2$ humidified incubator. We fixed the cells with 4% paraformaldehyde for 10 min and then stained them for TRAP expression following the manufacture's instruction at 0, 1, 3, and 5 days after inducible stimulation. For immunofluorescence (IF) staining, BMMs were washed, fixed in 4% paraformaldehyde, and permeabilized with 0.1% Triton X-100. We blocked the cells for 30 min at room temperature then incubated them with primary antibodies against RANK (1:100, ab13918, Abcam), TLR2 (1:100, mAb12276, Cell Signaling) over night at 4°C. Then, the cells were incubated with secondary antibodies and DAPI (1:250, H-1200, Vector) in the dark for 1 hr at room temperature. Photomicroscopic images were obtained by fluorescence microscopy (Olympus BX51, DP71) or confocal microscopy (Zeiss LSM 880) and analyzed by ImageJ software (National Institutes of Health, Bethesda). For pit formation assays, BMMs were seeded in a 96-well plate coated by the bone slice (IDS DT-1BON1000-96, Immunodiagnostic Systems Inc) at a density of $1 \times 10^4$ cells per well and stimulated with M-CSF (30 ng/ml) and RANKL (200 ng/ml) for 1 week. A 5% bleach solution was then added into the 96-well bone slice

surface to remove the cells. The bone resorption area was determined by Toluidine blue staining (T3260-5G, Sigma-Aldrich) following the standard protocol provided by the manufacture. To examine the function of TLR2 and sialylation for osteoclast fusion, BMMs were treated with either vehicle (PBS), the TLR2 inhibitor C29 (150 µM), or the sialyltransferase inhibitor soyasaponin Bb (10 mg/kg), along with M-CSF and RANKL for 3 days.

## Flow cytometry and cell sorting

Flow cytometry and cell sorting experiments were performed on CIA mice, DBA1/J, and WT mice. Hindlimbs were collected to flush the bone marrow from the femur and tibia with a fluorescence-activated cell sorting (FACS) buffer (5% FBS in PBS) and pipetted as a cell suspension. The cell suspension was filtered by a 40 µm strainer, and the cells were collected by centrifugation. The products were resuspended in ammonium-chloride potassium (ACK) lysing buffer (Quality Biological, Inc, Gaithersburg) for 10 min for the lysis of red blood cells. We resuspended the cells after lysis with 100 µl FACS buffer and incubated them with allophycocyanin (APC) anti-mouse RANK antibody (1:100, 119808, Biolegend) and Brilliant Violet 421 (BV421) anti-mouse TLR2 antibody (1:100, 565908, Biosciences) or Maackia amurensis lectin (1:100, B-1265–1, Vector Laboratories) for 30 min at 4°C. For the sialic acid test, we washed out the primary antibody using FACS buffer and further incubated the cells with FITC-conjugated Streptavidin for another 30 min at 4°C. RANK$^+$TLR2$^+$ and RANK$^+$TLR2$^-$ cells were sorted according to APC and BV421 fluorescence. FACS was performed using a three-laser cell sorter (BD FACSAria IIu Cell Sorter, BD Biosciences, San Jose) and analyzed with FACSDiva software (version VI, BD Biosciences). Flow cytometry was performed using a BD LSR II flow cytometer (BD Biosciences) and analyzed with FlowJo software (version 10, BD Bioscience).

## RT-qPCR

The total RNA was extracted from cultured cells using the RNA isolation RNeasy mini kit (Qiagen), according to the manufacturer's instructions. The purity of RNA was tested by the ratio of 260/280 nm. For qPCR, RNA was reverse transcribed into complementary DNA using PrimeScript RT Master Mix reagent kit and followed the protocol from the manufacture (Takara). Then qPCR was performed with SYBR Green-Master Mix (Qiagen, Hilden, Germany) on a C1000 Thermal Cycler (Bio-Rad Laboratories, Hercules). Relative expression was calculated for each gene by the $2^{-\Delta\Delta CT}$ method with *Gapdh* as the internal control for normalization. The primers used for each gene were as follows:

> *Gapdh* forward: 5'-CATCACTGCCACCCAGAAGACTG-3',
> *Gapdh* reverse: 5'-ATGCCAGTGAGCTTCCCGTTCAG-3'
> *Bmp4* forward: 5'-ATTCCTGGTAACCGAATGCTG-3'
> *Bmp4* reverse: 5'-CCGGTCTCAGGTATCAAACTAGC-3'
> *Bmi1* forward: 5'-AAATCCCCACTTAATGTGTGTCC-3'
> *Bmi1* reverse: 5'-CTTGCTGGTCTCCAAGTAACG-3'
> *Maf* forward: 5'-GGAGACCGACCGCATCATC-3'
> *Maf* reverse: 5'-TCATCCAGTAGTAGTCTTCCAGG-3'
> *Fgf2* forward F: 5'-AGAAGAGCGACCCTCACATCA-3'
> *Fgf2* reverse: 5'-CGGTTAGCACACACTCCTTTG-3'
> *Vegfa* forward: 5'-AGGGCAGAATCATCACGAAGT-3'
> *Vegfa* reverse: 5'-AGGGTCTCGATTGGATGGCA-3'
> *Ang* forward: 5'-CCAGGCCCGTTGTTCTTGAT-3'
> *Ang* reverse: 5'-GCAAACCATTCTCACAGGCAATA-3'
> *Csf1r* forward: 5'-GTGTCAGAACACTGTAGCCAC-3'
> *Csf1r* reverse: 5'-TCAAAGGCAATCTGGCATGAAG-3'
> *Grb2* forward: 5'-ACGGAGCCGGGAAGTATTTC-3'
> *Grb2* reverse: 5'-GGTTCCTGGACACGGATGTTG-3'
> *Fos* forward: 5'-CGGGTTTCAACGCCGACTA-3'
> *Fos* reverse: 5'-TGGCACTAGAGACGGACAGAT-3'
> *Nfkb1* forward: 5'-ATGGCAGACGATGATCCCTAC-3'
> *Nfkb1* reverse: 5'-CGGAATCGAAATCCCCTCTGTT-3'
> *Cd44* forward: 5'-TCGATTTGAATGTAACCTGCCG-3'
> *Cd44* reverse: 5'-CAGTCCGGGAGATACTGTAGC-3'

*Ccr2*forward: 5'-ATCCACGGCATACTATCAACATC-3'
*Ccr2* reverse: 5'-TCGTAGTCATACGGTGTGGTG-3'
*Ocstamp* forward: 5'-CTGTAACGAACTACTGACCCAGC-3'
*Ocstamp* reverse: 5'-CCAGGCTTAGGAAGACGAAGA-3'
*Cd47* forward: 5'-TGGTGGGAAACTACACTTGCG-3'
*Cd47* reverse: 5'-CGTGCGGTTTTTCAGCTCTAT-3'
*Ostm1* forward: 5'-GAGCTGACCGCCTGTATGG-3'
*Ostm1* reverse: 5'-ATGTTTCGGCTGATGTTGTCC-3'

## scRNA-seq analysis

Sample preparation and scRNA-seq: GemCode Single Cell platform (10 X genomics) was used to determine the transcriptome of single cells. A single-cell suspension was obtained from the flushed bone marrow mixture of the mice femur (n=3 mice from each sample group (CIA vehicle-treated, CIA soyasaponin Bb treated, and DBA vehicle control)) and subjected to Ficoll gradient centrifugation to enrich for mononuclear cells. Samples were profiled based on viability and amount and a range of 81–85% viability was achieved. A total of 10,000 cells were recorded for each run, followed by GEM-RT and cDNA amplification. Single-cell library preparation was performed by the Johns Hopkins School of Medicine for Genomics and Bioinformatics (Core facility) and the quality of cDNA samples was examined by an Agilent 2100 Expert High Sensitivity DNA Assay. Single-cell libraries were constructed by attaching P5 and P7 primer sites and sample indices to the cDNA. ScRNA-seq was performed on the Illumina NovaSeq 6000 S2 Flow Cell according to the manufacturer's protocol to a depth ranging from 315 to 377 million reads per sample. The data is available in GEO under accession code GSE221704.

Quality control, filtering, and downstream analysis of scRNA-seq data: CellRanger (version 5.0.1) was used to perform the initial processing of sequencing reads and alignment to the mm10 reference genome. Quality control was performed in Seurat (version 4.0.3) for each individual sample prior to concatenation and batch correction. Quality control was performed to remove doublets, broken cells/fragments, and dead or dying cells. Cells were selected with total counts less than the 95th percentile of total counts for that sample and less than 15% mitochondrial gene content. Features detected in fewer than five cells were excluded and cells expressing fewer than 200 features were excluded. Ribosomal genes, mitochondrial genes, and MALAT1 were removed from the feature list. Individual samples were log normalized and highly variable genes (HVGs) were computed using the variance stabilizing transformation method in Seurat. HVGs were computed for each individual sample and for all concatenated samples, and HVGs were selected as the union of all HVG lists. Individual samples were merged in Seurat and the resulting merged data were scaled and an estimation of the cell cycle phase was performed. Principal component analysis (PCA) was performed on the merged data and batch correction was performed using Harmony (version 0.1.0) to obtain corrected principal components. UMAP embeddings, K nearest neighbors, and Leiden clusters (resolution 1.0) were computed on Harmonized principal components. The number of Harmonized PCs used to calculate the embeddings (n=35) was determined by inspection of the variance ratio plots to determine which PCs accounted for the majority of variation in the data.

Cluster annotation and differential gene expression analysis: Clusters were annotated based on marker gene expression, differential expression analysis on annotated clusters, and SingleCellNet classification. Differential gene expression analysis was performed in Seurat using the FindMarkers() function and Poisson method. The threshold for minimum percentage expression of a gene within a given cluster was set to 0.1 and the threshold for log fold change was set to 0.25. p-values were adjusted using p.adjust() function in R and results with FDR-adjusted p-value <0.05 were used for cluster annotation. SingleCellNet version 0.1.0 (**Tan and Cahan, 2019a**) was used to classify single cells in our data by training a top-pair Random Forest classifier on a well-annotated reference dataset of the mouse bone marrow (The Tabula Muris Consortium). The performance of the trained classifier for each cell type was assessed using a precision-recall curve prior to the classification of query data. Classification results were visualized using an attribution plot to depict the composition of each training data cell type within query data categories of interest (https://pcahan1.github.io/singleCellNet/; **Tan and Cahan, 2019b**). Following annotation, subpopulations of interest (i.e. RANK+ monocytes and macrophages) were assigned to subsets from the larger dataset and the analysis steps above were

repeated for each subpopulation. Plots of gene expression were made in Seurat using the DotPlot(), ViolinPlot(), and FeaturePlot() functions.

Slingshot pseudotime and dynamic gene regulatory network reconstruction: Slingshot pseudotime was computed on harmonized principal components for subpopulations of interest (RANK⁻, TRAP⁻, and/or TLR2⁻ expressing monocytes/macrophages and RANK⁺ monocytes/macrophages, respectively) (*Street et al., 2018*). Macrophages were selected as the end cluster. Pseudotime values were inputted into Epoch (version 0.0.0.9000) for dynamic gene regulatory network (GRN) reconstruction. Epoch was used to identify genes that were significantly dynamically expressed (p<0.05) and GRN reconstruction was performed using the Pearson method with zThresh = 2 (*Su et al., 2022*). Epochs were defined using pseudotime to identify n=2 epochs, or pseudotemporal periods, per lineage. Differential network analysis was performed to compare dynamic GRNs from RANK⁺TLR2⁺ monocytes/macrophages and RANK⁺TLR2⁻ monocytes/macrophages with thresholds of 4.0 for differentially active edges and 7.0 for consideration of an edge in a given network. Plots of dynamic GRNs, top regulators and their targets, and differential networks were made in Epoch (https://github.com/pcahan1/epoch; *Cahan Lab, 2022*).

## CHIP assay

Primary BMMs were isolated for CHIP assay according to the instruction from the Pierce Agarose CHIP Kit (Cat. 26156, Thermo Fisher) using a CHIP-grade antibody to c-Fos (mAb2250, Cell Signaling). In brief, we crosslinked the cell pellet using Glycine Solution after fixation in 1% formaldehyde. The cells were lysed in membrane extraction lysis buffer and nuclear extraction lysis buffer, along with MNase digestion (DTT, MNase Digestion Buffer). Of the sample, 10% was removed as an input control. CHIP was performed according to the protocol provided by manufacturers using an antibody to c-Fos. Anti-RNA polymerase II and control IgG were used as positive and negative controls, respectively. The purified DNA was then analyzed by PCR and electrophoresis.

## Behavioral tests

Von Frey tests were performed to evaluate the response to a mechanical stimulus. The mice were placed on a metal wire plate with a clear plastic cage and were acclimatized for 1 hr before the test. The von Frey filament (0.4 g force, BIOSEB) was applied to the plantar surface of each hind paw continuously for three seconds or till paw withdrawal. The frequency of paw withdrawal was recorded during a total of ten applications, which included 2 s intervals between each repeat. The test was repeated at least three times for each mouse and the average frequency was processed for further analysis.

*Hargreaves tests* were used on mice to examine the analgesia level. Groups of mice were utilized for the analgesia test, and they were allowed to acclimatize for 1 hr before the experiment. When animals were ready, the radiant heat produced by a strong light beam (IITC Life Science Inc) was applied to the plantar surface of each hind paw until they flicked the paw. The time was recorded as the latency time response to heat-producing light beam stimulation. The test was repeated at least three times for each mouse and the average latency was processed for further analysis.

## Statistics

We used SPSS 22.0 (IBM Corp., Armonk, NY) to perform the statistical analysis for all data. And we presented the results as mean ± standard deviations. The one-way ANOVA with Tukey's multiple comparisons test was used to analyze data among multiple groups. And an unpaired, two-tailed Student's t-test was used to compare data between the two groups. For the scRNA-seq data, the Poisson test using the FindMarkers() function in Seurat for dot plot data, with FDR adjustment using p.adjust() in R for violin plot data for statistical analysis. For all experiments, p<0.05 was considered to be significant.

# Acknowledgements

This research was supported by NIH National Institute on Aging under Award Number R01 AG076783 R01 AG068997 and P01 AG066603 (to XC).

# Additional information

## Competing interests

Mei Wan: Reviewing editor, *eLife*. The other authors declare that no competing interests exist.

## Funding

| Funder | Grant reference number | Author |
|---|---|---|
| National Institute on Aging | R01 AG076783 | Xu Cao |
| National Institute on Aging | R01 AG068997 | Xu Cao |
| National Institute on Aging | P01 AG066603 | Xu Cao |

The funders had no role in study design, data collection and interpretation, or the decision to submit the work for publication.

## Author contributions

Weixin Zhang, Conceptualization, Investigation, Methodology, Writing – original draft; Kathleen Noller, Investigation, Methodology; Janet Crane, Mei Wan, Investigation; Xiaojun Wu, She made an important contribution to manuscript revising; Patrick Cahan, Conceptualization; Xu Cao, Conceptualization, Funding acquisition, Writing – review and editing

## Author ORCIDs

Weixin Zhang (ID) http://orcid.org/0000-0002-1009-101X
Kathleen Noller (ID) http://orcid.org/0000-0001-7915-3269
Mei Wan (ID) http://orcid.org/0000-0001-9404-540X
Xu Cao (ID) http://orcid.org/0000-0001-8614-6059

## Ethics

All animals were kept in the Animal Facility of the Johns Hopkins University School of Medicine. The animal protocol was approved by the Institutional Animal Care and Use Committee of Johns Hopkins University, Baltimore, MD, USA (MO21M276).

## Decision letter and Author response

Decision letter https://doi.org/10.7554/eLife.85553.sa1
Author response https://doi.org/10.7554/eLife.85553.sa2

# Additional files

## Supplementary files

• MDAR checklist

## Data availability

Sequencing data have been deposited in GEO under accession codes GSE221704.

The following dataset was generated:

| Author(s) | Year | Dataset title | Dataset URL | Database and Identifier |
|---|---|---|---|---|
| Zhang M, Noller K, Crane J, Wan M, Cahan P, Cao X | 2022 | RANK+TLR2+ Myeloid Subpopulation Converts Autoimmune to Joint Destruction in Rheumatoid Arthritis | https://www.ncbi.nlm.nih.gov/geo/query/acc.cgi?acc=GSE221704 | NCBI Gene Expression Omnibus, GSE221704 |

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
