## [Editor Report]

New cell populations of RANK^+^TLR2- cells and RANK^+^TLR2+ cells have been identified in RA mice by scRNA-Seq method. The RANK^+^TLR2- cells could differentiate into a TRAP+ osteoclast lineage but did not fuse to osteoclasts, while RANK^+^ TLR2+ myeloid monocyte population plays an important role in fusion and osteoclast formation. These findings, with other results, provide novel information on progression of RA disease.

---

## [Decision Letter]

**Decision letter after peer review:**

Thank you for submitting your article "RANK^+^TLR2+ Myeloid Subpopulation Converts Autoimmune to Joint Destruction in Rheumatoid Arthritis" for consideration by *eLife*. Your article has been reviewed by 3 peer reviewers, one of whom is a member of our Board of Reviewing Editors, and the evaluation has been overseen by Mone Zaidi as the Senior Editor. The following individual involved in review of your submission has agreed to reveal their identity: Ting-Yu Wang (Reviewer #3).

Essential revisions:

1) The authors need to determine if Maf could regulate its downstream target gene in RANK^+^ TLR2- cells.

2) The authors need to explain why the percentage of RANK^+^TLR2+ cells increase on day5 but decrease on day3.

3) The authors need to explain why they choose female mice when they develop CIA-induced RA model.

*Reviewer #1:*

In this study, the authors investigated the molecular mechanisms of autoimmune-induced joint damage during rheumatoid arthritis (RA) progression. Using single cell RNA-sequencing method and CIA-induced RA model. They identified the RANK^+^ TLR2- subset myeloid monocyte, which could differentiate into TRAP+ osteoclast lineage cells but did not fuse to osteoclasts. The type of cells acquired an anabolic phenotype with enriched expression of pro-angiogenic and pro-osteogenic growth factors. In contrast, the RANK^+^ TLR2+ myeloid monocyte and its sialylation in the RA mouse could mediate the transition from autoimmunity to osteoclast fusion and bone resorption. The inhibition of sialyltransferases or treatment with a TLR2 inhibitor blocked osteoclast fusion and rescued bone resorption.

Overall speaking, this is a well-designed and carefully performed study with novel information generated. The data obtained from this study, in general, support their conclusion.

The data presented in this study are novel and interesting. However, few questions and comments need to be addressed.

It would be interesting to know:

1) how RANK^+^ TLR2+ cells and RANK^+^ TLR2+ cells respond to TNF-α.

2) if Maf could regulate its downstream target gene in RANK^+^ TLR2- cells.

3) if M-CSF/RANKL could induce osteoclast fusion for longer time culture period, such as 10 or 14 days in RANK^+^ TLR2- cells.

*Reviewer #2:*

This study also presents a therapeutic target for preventing autoimmune-mediated joint destruction and promoting bone anabolism in various autoimmune diseases and conditions, particularly rheumatoid arthritis. In this manuscript, authors uncovered that autoimmune disease converted to osteoclast joint destruction as a bone disease. Specifically, preosteoclasts identified as RANK^+^TLR2+ monocyte/macrophages are fused to form osteoclasts. In rheumatoid arthritis, RANK^+^TLR2+ monocyte/macrophages number are significantly increased. As a result, osteoclast bone resorption is elevated to destruct joints. Furthermore, the characteristics of this population were well demonstrated in the results by both single cell sequence technic and in vitro experiments. Overall, the experiments are logically designed and well executed. The manuscript is clearly written.

I have the following specific questions:

1. In Figure1. G, the percentage of RANK^+^TLR2+ positive cells increased on day 5 while it decreased on day 3. What is the explanation?

2. In Figure2. A, there are several dots in the RANK^+^TLR2- group which are obviously smaller than them in the RNAK^+^TLR2+ group, however, some of them did not show statistic difference. Please provide the specific statistical method for the calculation.

3. In Figure3. E, it lacks a specific label for RANK^+^TLR2- population. It is difficult to see that TLR2 negative population but not TLR2 positive population cannot fuse to multinuclear osteoclast.

4. In Figure4. F, the images of TRAP staining only provide a limited view of the knee joint. Please provide the larger area to show the difference between groups.

5. In Figure7. O-U, the authors evaluated the function of osteoclast by qPCR experiment. However, the genes chosen were different from them in Figure5 F-L. The expression of CCR2, CSF1R genes in Figure5 experiments, and the expression of OCSTAMP, OSTM1 genes in Figure7 experiment will be helpful.

*Reviewer #3:*

This landmark study presents a valuable finding on the transition from autoimmune to joint destruction in rheumatoid arthritis. Authors had shown sufficient molecular results which well supported results in vivo. The whole paper has strong logic, careful experimental design and innovation. The conclusions of this paper are well supported by data. The work will be of broad interest to bone biologists and immunologists.

1. In the manuscript, female DBA/1J mice were chosen to induce CIA model. Please specify the reasons to choose the gender of the mice. Whether the period and estrogen level of the female mice have impacts on the results and the conclusions of the paper?

2. Inflammation is very important in RA. Whether inhibition of TLR2 sialylation has an influence on the inflammation of RA?

3. Samples from RA patients and cell-specific gene knock-out mice may strengthen the conclusions in the paper.

---

## [Author Response]

Reviewer #1:In this study, the authors investigated the molecular mechanisms of autoimmune-induced joint damage during rheumatoid arthritis (RA) progression. Using single cell RNA-sequencing method and CIA-induced RA model. They identified the RANK^+^ TLR2- subset myeloid monocyte, which could differentiate into TRAP+ osteoclast lineage cells but did not fuse to osteoclasts. The type of cells acquired an anabolic phenotype with enriched expression of pro-angiogenic and pro-osteogenic growth factors. In contrast, the RANK^+^ TLR2+ myeloid monocyte and its sialylation in the RA mouse could mediate the transition from autoimmunity to osteoclast fusion and bone resorption. The inhibition of sialyltransferases or treatment with a TLR2 inhibitor blocked osteoclast fusion and rescued bone resorption.Overall speaking, this is a well-designed and carefully performed study with novel information generated. The data obtained from this study, in general, support their conclusion.The data presented in this study are novel and interesting. However, few questions and comments need to be addressed.It would be interesting to know:1) how RANK^+^ TLR2+ cells and RANK^+^ TLR2+ cells respond to TNF-α.

To address this question, we sorted RANK^+^TLR2^-^ and RANK^+^TLR2^+^ populations, respectively. The sorted cells were treated with RANKL (200ng/ml) and M-CSF (30ng/ml) with recombinant mouse TNF-α (20ng/ml) or vehicle for three days. The osteoclast differentiation was determined by TRAP staining. The results demonstrated that TNF-α significantly promotes the osteoclastogenesis of RANK^+^TLR2^+^ population, while it had no effect on osteoclast differentiation of RANK^+^TLR2^-^ population relative to vehicle treatment (see Author response image 1).

**Author response image 1. sa2fig1:** The representative images for TRAP staining of RANK^+^TLR2+ and RANK^+^TLR2- cells with RANK (200ng/ml) and M-CSF (30ng/ml) and vehicle or TNF-α (20ng/ml) treatment for three days. Scale bar, 0.1 mm.

2) if Maf could regulate its downstream target gene in RANK^+^ TLR2- cells.

Yes, in our manuscript, we have demonstrated that the expression of *Maf* downstream genes including *Cd74, Sub1, Ms4a7*, and *Fcgr2b* were regulated in RANK^+^TLR2^-^ myeloid cells. All of these target genes are related to myeloid cell/macrophage differentiation. Particularly, our analysis showed that expression of the myeloid cell marker gene *Cd74* was activated by transcription factors *Maf* and *Mafb* in RANK^+^TLR2^-^ myeloid cells (Figure. 2C, D).

3) if M-CSF/RANKL could induce osteoclast fusion for longer time culture period, such as 10 or 14 days in RANK^+^ TLR2- cells.

To address the question, we sorted RANK^+^TLR2^-^ cells and cultured for 10 days and we did not observe osteoclast fusion and differentiation in TRAP staining as shown in Author response image 2.

**Author response image 2. sa2fig2:** RANK^+^TLR2- cell were cultured for 10 days with RANK(200ng/ml) and M-CSF(30ng/ml) and stained by TRAP. Scale bar, 0.1 mm.

Reviewer #2:This study also presents a therapeutic target for preventing autoimmune-mediated joint destruction and promoting bone anabolism in various autoimmune diseases and conditions, particularly rheumatoid arthritis. In this manuscript, authors uncovered that autoimmune disease converted to osteoclast joint destruction as a bone disease. Specifically, preosteoclasts identified as RANK^+^TLR2+ monocyte/macrophages are fused to form osteoclasts. In rheumatoid arthritis, RANK^+^TLR2+ monocyte/macrophages number are significantly increased. As a result, osteoclast bone resorption is elevated to destruct joints. Furthermore, the characteristics of this population were well demonstrated in the results by both single cell sequence technic and in vitro experiments. Overall, the experiments are logically designed and well executed. The manuscript is clearly written.I have the following specific questions:1. In Figure1. G, the percentage of RANK^+^TLR2+ positive cells increased on day 5 while it decreased on day 3. What is the explanation?

When RANK^+^TLR2^+^ cells were undergone cell-cell fusion for formation multinuclear osteoclasts on day 3, the expression level of RANK and TLR2 were decreasing in the fusion osteoclasts. However, some of the mononuclear RANK^-^TLR2^-^ cells under stimulation of RANKL and M-CSF stimulation differentiated to RANK^+^TLR2^+^ cells at day 5. Besides, the mature osteoclast occupied a large area in one view, so, it appears that the percentage of RANK^+^TLR2^+^ cells in one limited view increased on day 5.

2. In Figure2. A, there are several dots in the RANK^+^TLR2- group which are obviously smaller than them in the RNAK^+^TLR2+ group, however, some of them did not show statistic difference. Please provide the specific statistical method for the calculation.

In the revised manuscript, the statistical method was explained. Specifically, differential gene expression was performed in Seurat using the FindMarkers function using the Poisson method, and asterisks in Figure2. A represent gene expression differences with FDR-adjusted *p* value < 0.05. FDR adjustment of *p* values was performed using the p.adjust() function from the R base *stats* package. Genes which do not have a corresponding asterisk do not have a significant difference in expression between RANK^+^TLR^-^ and RANK^+^TLR^+^ groups.

3. In Figure3. E, it lacks a specific label for RANK^+^TLR2- population. It is difficult to see that TLR2 negative population but not TLR2 positive population cannot fuse to multinuclear osteoclast.

Sorry for missing the label. The label has been added in the revised manuscript.

4. In Figure4. F, the images of TRAP staining only provide a limited view of the knee joint. Please provide the larger area to show the difference between groups.

As suggested, a larger view of the representative images have been added in the revised manuscript in Figure. 4F.

5. In Figure7. O-U, the authors evaluated the function of osteoclast by qPCR experiment. However, the genes chosen were different from them in Figure5 F-L. The expression of CCR2, CSF1R genes in Figure5 experiments, and the expression of OCSTAMP, OSTM1 genes in Figure7 experiment will be helpful.

As suggested, the expression of *CCR2, CSF1R, OCSTAMP, OSTM1* genes were examined by qPCR. The results are shown in Author response image 3.

**Author response image 3. sa2fig3:** Expression of osteoclast genes in RANK^+^TLR2^+^ cells with soyasaponin Bb treatment. (A-B) The mRNA expression of CCR2, CSF1R genes in RANK^+^TLR2+ monocytes treated with PBS, C29, or soyasaponin Bb, respectively (n = 3, one-way ANOVA with Tukey’s multiple comparisons test). (C-D) The mRNA expression of OCSTAMP, OSTM1 genes in RANK^+^TLR2+ monocytes treated with in CIA mice treated with PBS or soyasaponin Bb relative to DBA mice (n = 3, one-way ANOVA with Tukey’s multiple comparisons test). All data are means ± SD. *P < 0.05, ***P < 0.001, ****P < 0.0001.

Reviewer #3:This landmark study presents a valuable finding on the transition from autoimmune to joint destruction in rheumatoid arthritis. Authors had shown sufficient molecular results which well supported results in vivo. The whole paper has strong logic, careful experimental design and innovation. The conclusions of this paper are well supported by data. The work will be of broad interest to bone biologists and immunologists.1. In the manuscript, female DBA/1J mice were chosen to induce CIA model. Please specify the reasons to choose the gender of the mice. Whether the period and estrogen level of the female mice have impacts on the results and the conclusions of the paper?

We appreciate the question. Epidemiological study showed that the risk of suffer rheumatoid arthritis is almost nine times higher in women than in men. CIA female mouse model mimics the phenotype of human rheumatoid arthritis. We do not have evidence to show whether period or estrogen level have impacts on the phenotype of RA in CIA female mouse model.

2. Inflammation is very important in RA. Whether inhibition of TLR2 sialylation has an influence on the inflammation of RA?

Yes, we do have a clinical score result to demonstrate the inflammation phenotype after inhibition of TLR2 sialylation. The clinical score significantly decreased in soyasaponin Bb treated mice relative to vehicle treated group (Figure7. H). It indicated that inhibiting TLR2 sialylation could ameliorate paw swelling and thickness.

3. Samples from RA patients and cell-specific gene knock-out mice may strengthen the conclusions in the paper.

We are applying an IRB for the clinic sample collection in the future and patient samples will be used to validate the results from knock-out mice.